

# Probabilistic seismic hazard analysis using logic tree approach-Patna District (India)

Panjamani Anbazhagan[1], Ketan Bajaj[1], Karanpreet Matharu[1], Sayed S. R. Moustafa[2], Nassir S. N. Al-Arifi[2]

[1]Department of Civil Engineering, Indian Institute of Science, Bangalore, India.
[2]Geological and Geophysics Department, King Saud University, Riyadh, Saudi Arabia

*Correspondence to*: P. Anbazhagan (anbazhagan@iisc.ac.in)

**Abstract.** PGA and SA distribution for Patna district is presented considering both classical and zoneless approach through logic tree frame work to capture the epistemic uncertainty. Seismicity parameters are calculated by considering completed
and mixed earthquake data. Maximum magnitude was calculated using three methods namely incremental method, Kijko method and regional rupture characteristics approach. Best suitable GMPE was selected by carrying out "efficacy test" using log likelihood. Uniform hazard response spectra have been compared with Indian standard BIS 1893. PGA varies from 0.38 g to 0.30 g from southern to northern periphery considering 2 % probability of exceedence in 50 years.

## 1 Introduction

Seismic hazard analysis is effective in presenting the potentially damaging phenomenon associated with earthquake. Earthquake disaster is not only associated with collapsing of structures due to ground shaking but also triggers fire, liquefaction, landslide, and Tsunami. So, it is indispensable to forecast the ground shaking level to serve the engineering needs in mitigating the risk associated with earthquakes. In India, moderate earthquakes ($M_w < 7$) including Anjar 1956, Koyna 1967, Udaypur 1988, Uttarkashi 1991, Chamoli 1999 have caused significant damage in last 10 decades (Nath and
Thingbaijam 2012). Besides, many great events (2015, Nepal earthquake) have originated from Subduction zone. The seismic gap and thick soft soil sediments makes the scenario more dangerous for cities close to Himalayan region. Apart from this, improper planning, increase in population density, poor land use and substandard construction practices in these cities magnify the prevailing seismic risk. Most of the existing seismic hazard maps are mainly on macro level for different parts of Indian subcontinent and are not up to state of art knowledge in engineering seismology. For example, Khattri et al.
(1984) developed a hazard map representing peak ground acceleration (PGA) of entire India with 10% probability of exceedence in 50 years. Under the Global Seismic Hazard Assessment Program (GSHAP), Bhatia et al. (1999) presented a probabilistic seismic hazard analysis (PSHA) of India framework. Mahajan et al. (2010) delivered PSHA for the northwestern Himalayas. Recently, National Disaster Management Authority (NDMA 2010) and Nath and Thingbaijam (2012) have given the PSHA map for entire India. In addition, Kumar et al. (2013) has developed a deterministic seismic
hazard analysis (DSHA) and PSHA map for Lucknow region considering local and active seismic gap. Additionally, the



current Indian Standard (IS 1893 2016) code consists of many constraints such as poor delamination of active seismic sources, lack of vulnerable sources study, improper seismic hazard parameters which are not region-specific and limited soil amplification consideration (Anbazhagan et al. 2014). Subsequently an updated seismic hazard map at micro level is essential for cities near to the Himalayan region, by considering new data, updated knowledge and improvement in previous methodologies.

There are two types of uncertainties associated with hazard analysis. One is due to randomness of the nature of earthquake and ground motion prediction named as aleatory uncertainty while other is due to incomplete knowledge of earthquake process named as epistemic uncertainty. Former can be easily reduced by integrating the distribution of ground motion about the median (Bommer and Abrahamson 2006) and latter can be assessed using logic tree approach. In this study, logic tree framework has been made to reduce the epistemic uncertainty in the final hazard value calculation. Epistemic uncertainty is due to improper knowledge about the process involve in earthquake events and algorithms used to model them. Generally, ground motion prediction models are more representative when the appropriate region-specific models of wave propagation are not available. This can be examined by incorporation of logic tree in the hazard analysis study. Logic tree represents the various nodes that defines alternative input choices and each branch is assigned a weight that signifies the quantitatively or qualitatively degree of plausibility assigned. To quantify the epistemic uncertainty, different branches of logic tree need to be considered which is based on source models, regionalization of $b - value$, determination of magnitude of completeness and maximum magnitude and epistemic uncertainty in GMPE using the representative suitable approach.

In the present study, PSHA of Patna district (India) at micro level has been prepared along with the response spectrum by reducing epistemic uncertainty. Patna lies at 250 km from the Central Seismic Gap (Khattri 1987) in Himalayan region where the huge devastation and destruction due to 1803, 1934 Bihar-Nepal and 2015 Nepal earthquakes were reported. As per Bilham (2015), a large earthquake appears to be imminent in future due to failure of rupturing of the main fault beneath the Himalaya because of Nepal 2015 earthquake. Hence such studies need to be done for the cities that lie within the vicinity of Himalayan region and on Indo Gangetic Basin. Seismic sources and seismic events have been taken for 500 km radius around the district centre as per Anbazhagan et al. (2015a). The '$a$' and '$b$' parameters have been arrived by taking into consideration the completed earthquake data using Gutenberg-Richter (G-R) relationship and mixed data using methods proposed by Woessner and Wiemer (2005). The magnitude of completeness ($M_c$) is also calculated by nine methods proposed by Woessner and Wiemer (2005). Maximum magnitude has been determined weighted mean using increment factor on maximum observed magnitude, Kijko and Sellevoll (1989) and regional rupture characteristics (Anbazhagan et al. 2015b). Ground motion prediction equations (GMPEs) has been selected from the twenty-seven numbers of applicable GMPEs for the region. The seismic hazard map for Patna district has been developed using PSHA applying probabilistic methods viz. classical method proposed by Cornell (1968) which was later upgraded by Algermissen et al. (1982) and smoothed-gridded seismicity models using areal source and four models proposed by Frankel (1995). For the development of hazard map using areal approach, delineation of seismic zones has been done based on the seismicity parameters i.e. '$a$', '$b$' and $M_c$. The hazard curves between mean annual rate of exceedence versus PGA and spectral acceleration ($S_a$) are



developed at the rock levels by both models. The final hazard map in terms of the rock level peak ground acceleration values are mapped for 2% and 10% probability of exceedence in 50 years i.e. return period of 2475 and 475 years based on logic tree. Additionally, hazard map for $S_a$ at 0.2 and 1 s for return period of 2475 and 475 years is also given. Furthermore, uniform hazard spectrum for Patna district at rock level for return period of 2475 and 475 years based on logic tree has been

estimated and compared with Indian standard IS 1893.

## 2 Geology, Seismotectonics and seismicity of study area (SA)

Regional seismicity, geological, seismological and seismotectonics information of SSA have been assembled and evaluated for a desirable radius for seismic hazard analysis. The present study area has longitude 85.144°E and latitude 25.611°N and is near to various rivers such as Gandak in west, Ganga in southern side, Kosi and Bhagmati rivers in north side. Patna lies in

the Seismic zone IV with zone factor of 0.24 as per IS: 1893 (2016). To carry out a seismic hazard analysis, details and documentation about seismic features such as faults, shear zones and lineaments along with all earthquake events ($M_w > 4$) that have occurred in the SSA are mandatory. Based on damage distribution map i.e. isoseismal map (1833 Nepal earthquake and 1934 Bihar-Nepal earthquake) and location of Main Boundary Trust, Main Central Trust and Himalayan Frontal Thrust (HFT), a radius of 500 km has been selected for present SSA. The detail study about selecting SA of 500 km is given in

Anbazhagan et al. (2015a). Patna district lies near to the seismically active Himalayan belt and on the deep deposits of the Indo-Gangetic basin (IGB). Present study area is also surrounded by various active ridges as Monghyr-Saharsa Ridge Fault many active tectonic features such as Munger-Saharsa-Ridge Fault, and active faults such as East Patna Fault or West Patna Fault. Historic earthquakes such has 1833 Bihar, 1934 Bihar-Nepal, 1988 Bihar-Nepal has affected Patna city as far as economic loss and loss of lives is concerned. Many other earthquakes that have occurred near Bihar-Nepal border also prove

to be devastating for Patna district. In addition to that, north side Patna is near East and West Patna fault. The frequency of seismic events on these faults are high (Valdiya 1976; Dasgupta et al. 1987). Besides SSA is also at 250 km from the Himalayan plate boundary. These plate boundaries were the source of major historic earthquakes. Considering the above seismic aspects, Patna district, can be acknowledged under a high seismic risk. Thus, in the present work, PSHA of Patna district has been carried out by considering all seismic sources and earthquake events by reducing epistemic uncertainty

using logic tree approach.

Geographical information of India demonstrates that approximately 60% of the land is highly susceptible to earthquakes (NDMA, 2010). The tectonic feature of SA has been compiled from the Seismotectonic Atlas (SEISAT, 2010) published by the Geological Survey of India (GSI, 2000).  The seismotectonic map was developed by considering 500 km radius from Patna district boundary by considering linear sources (faults and lineaments) from SEISAT and published literatures.

Separation of MBT and MCT has been done and all the faults along with MBT and MCT have also been numbered. Seismotectonic map for Patna District is shown in Figure 1.



The earthquake data is collected from various agencies such as National Earthquake Information Centre (NEIC), International Seismological Centre, Indian Meteorological Department (IMD), United State Geological Survey (USGS), Northern California Earthquake Data Centre (NCEDC), and GSI. A total of 2325 events have been compiled which are in different magnitude scale such as local magnitude, surface wave magnitude and body wave magnitudes. To attain

uniformity, all the reported events are converted to moment magnitude ($M_w$) using relations given by Scordilis (2006) considering worldwide data. Furthermore, declustering algorithm proposed by Gardner and Knopoff (1974), modified by Uhrhammer (1986) was used for the separation of main event from dependent events. Out of 2325 events, 54% were noticed as dependent events i.e. 1272 events were documented as main shock for Patna region. The complete catalogue contains 454 events having moment magnitude less than 4 and 818 events with $M_w \geq 4$. To develop the seismotectonic map, the linear

source map was superimposed with the declustered earthquake events with and given as Figure 1. Near to MBT and MCT, earthquake events are densely located (See Figure 1) as compared to other part of seismotectonic map. As per Cornell (1968) and Frankel (1995) seismic study area need to be divided based on the seismicity or tectonic provision for calculating the significant hazard value from any potential source. Based on the event distribution SSA is divided into Region I (which belongs to MBT and MCT) and Region II. These regions were separated using a polygon, as shown in Figure 1; Region I fit

in to events inside and Region II belongs to events outside the polygon. Region I contained 280 events with $M_w$ 4 to 5, 197 events with $M_w$ 5.1 to 6, 26 events with $M_w$ 6.1 to 7 and 4 events with $M_w$ greater than 7, whereas region II contained a total of 310 significant events viz. 168 events with $M_w$ 4 to 5, 121 events with $M_w$ 5.1 to 6 and 21 events with $M_w$ 6.1 to 7. Both the regions were separately analysed for the seismic hazard estimation.

## 3 Seismicity Parameters

### 3.1 'a' and 'b' parameters

The most widely known Guttenberg-Richter (G-R) relationship (Gutenberg and Richter 1956) are usually used for the determination of '$a$' and '$b$' parameters for any SSA. The seismic recurrence rate can be precisely calculated only for the complete seismic event data. Stepp (1972) is used for examining the completeness of both the regions. Based on the analysis, it has been observed that for $M_w > 5$, catalogue is completed for 110 years for both the regions. However, for $M_w < 5$,

catalogue is completed for last 80 years and 70 years respectively for region I and region II. After determining the completeness of catalogue, G-R recurrence law for both the region has been estimated. The 'b' value for the region I and region II respectively were found as 0.87 and 0.97. Whereas the '$a$' value for region I and region II respectively for present study was determined as 5.32 and 4.98. More details about period of completeness and G-R recurrence law were described in Anbazhagan et al. (2015a).





## 3.2 Magnitude of completeness (M$_c$)

Magnitude of completeness is defined as the lowest magnitude at which 100% of the events in a space–time volume is detected (Rydelek and Sacks 1989; Taylor et al. 1990; Wiemer and Wyss 2000). $M_c$ is also important for mapping out seismicity parameters such as b-value of Gutenberg-Richter relationship. The magnitude of completeness was calculated using nine different methods defined by Woessner and Wiemer (2005). Addition to magnitude of completeness, these methods also estimate G-R '$a$', and '$b$' parameters. These methods are Maximum Curvature Method (M1), Fixed Minimum Magnitude observed ($M_{min}$) (M2), goodness of fit $M_{min}90$ (M3) and $M_{min}95$ (M4), Best combination of $M_{min}90$ and $M_{min}90$ and maximum curvature (M5), entire magnitude range (M6), Shi and Bolt (1982) method (M7), Bootstrap method (M8), Cao and Gao (2002) method (M9). Magnitude of completeness for Patna site for Region I and Region II (shown in Figure 1) was estimated using software package ZMAP (Wiemer, 2001), a MATLAB based programme. The '$a$' , '$b$' and $M_c$ from each method is represented as Figure 2 for method M1, M2, M3, M4, M5, M6, M7, M8 and M9 for both the regions. It has been observed that $M_c$ varies from 1.7 to 5.0 $M_w$ for region I and 1.9 to 4.9 $M_w$ for region II. So, for the further analysis, magnitude moment of 4.5 would be considered as magnitude of completeness. It is also observed that at R-value of 95% fit for the observed magnitude-frequency distribution cannot be modeled by a straight line for the region II due to lack of large amount of data. The Guttenberg-Richter '$a$' and '$b$' parameter calculated using these 9 methods is different from calculated using completed data with G-R relationship values for both the region. Calculated values of G-R '$a$' and '$b$' parameter for both the regions is given in Table 1. The value of '$a$' parameter calculated from the above methods vary from 3.11 to 6.57 for region I and 3.07 to 6.4 for region II. However, '$b$' parameter calculated from the above methods varies from 0.149 to 0.843 for region I and 0.176 to 0.848 for region II. The difference in '$a$' and '$b$' parameters determined using the above methods, as it is calculated based on magnitude of completeness using mixed data (Woessner and Wiemer 2005) instead of period of completeness for completed data of earthquake events. It has been seen from Table 1 that average value of '$a$'-parameter is 4.95 for region I which is low as compared with the number of earthquakes in the region. Similarly, average '$b$'-value of 0.522 and 0.661 for region I and region II is also low when compared to the number of earthquake events having larger magnitude. So, as per Woessner and Wiemer (2005), M6 method is capable for $M_c$ calculation as it synthetically maximises the available data and stabilises the $M_c$ value. Therefore, for further analysis, '$a$' and '$b$' value of 6.57 and 0.843 and 6.22 and 0.815 respectively had considered for region I and II. For further study, weight factor of 0.5 was given to each of the method (i.e. period of completeness and magnitude of completeness viz. M6) used to determine the 'a' and 'b' value for both the regions. The final value of 5.0 $M_w$ and 4.8 $M_w$ is adopted as magnitude of completeness for region I and II respectively for further study.

## 3.3 Maximum Magnitude estimation (M$_{max}$)

The maximum probable earthquake magnitude has been calculated using both deterministic and probabilistic approach. Three methods viz. conventional methods of increment of 0.5 in maximum observed magnitude ($M_{obs}^{max}$) based on 'b' values,

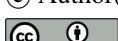



Kijko Method (Kijko and Sellevoll 1989) and regional rupture characteristics (Anbazhagan et al. 2015b) have been used in $M_{max}$ calculation. For the estimation of $M_{max}$ using Kijko and Sellevoll (1989), calculation of $M_c$ is already discussed above. Secondly, $M_{max}$ magnitude has been calculated by adding a constant value of 0.5 to $M_{max}^{obs}$ value at each fault (see Figure 1) like NDMA (2010) report. $M_{max}$ is also estimated using regional rupture characteristics by considering the $M_{max}^{obs}$

and possible seismic source. The whole procedure to calculate region-specific rupture characteristic was presented in Anbazhagan et al. (2015a). As per Risk Engineering Inc (1988) and others, increment varies from source zone to source zone and as per Wheeler (2009) short historical records produce samples of seismicity that are too small to constrain $M_{max}$. As per Anbazhagan et al. (2015b), $M_{max}$ estimated from probabilistic method i.e. Kijko and Sellevoll (1989) is sensitive to SSA and seismicity parameters of a region. However, $M_{max}$ determined using regional rupture characteristic is more reliable as it

depends upon the seismic source and rupture length. Taking these points into consideration a qualitative weight factor of 0.3, 0.3 and 0.4 has been assigned to incremental method, Kijko method and regional rupture method respectively. More weight is given to regional rupture approach as it accounts for rupture of seismic source which in turn depends upon the energy released for an event. Maximum magnitude calculating corresponding to each fault is submitted as an electronic material (Table S1) and available Anbazhagan et al. (2015a).

**4 Selection of Ground Motion Prediction Equation (GMPE)**

GMPEs has been selected based on the efficacy test recommended by Scherbaum et al. (2009) and Delavaud et al. (2009). There are various GMPEs are available for the active crustal region and basin. Out of various GMPEs, 27 GMPEs are applicable for the present SA. The details of the efficacy test have been given in Anbazhagan et al. (2015c). Detail of these GMPEs are given in Anbazhagan et al. (2015a). Similar to Anbazhagan et al. (2015 a), the hypocentral distance is divided

into three length bins viz. 0-100 km, 100-300 km and 300-500 km. The determined PGA values are used to estimate the log-likelihood (LLH) values, further Data Support Index (DSI) given by Delavuad et al. (2012) is used to rank the best suitable GMPEs. Positive DSI values have been identified for each segment and ranked based on maximum to minimum values. Positive DSI values for Patna earthquake is marked as bold in Table 3. It has been seen from Table 3 that three GMPE such as ANBU-13, NDMA-10 and KANO-06 can be used up to 100 km of hypocentral distance. For 100-300 km distance,

ANBU-13, NDMA-10, KANO-06 and BOAT-10 and for hypocentral distance greater than 300 km, NDMA-10 will be used for further hazard analysis. Seismic hazard values in terms of PGA and SA can be calculated considering these equations for each seismic source. In addition to that, LLH based weight as per Delavaud et al. (2012) for selected GMPEs were also calculated. The weight factors of 0.72, 0.17 and 0.11 are assigned with ANBU-13, NDMA-10 and KANO-06 up to 100 km of hypocentral distance. For 100-300 km distance, KANO-06, ANBU-13, NDMA-10 and BOAT-10 with weight factor of

0.32, 0.28, 0.26 and 0.14 are used and hypocentral distance greater than 300 km weight factor of 1 has been associated with NDMA-10. It can be noted here that only one GMPE is surfaced with positive DSI for distance segment of 300 km to 500 km and required additional GMPEs in this range, which is important for the far filed damage scenario in the region. These



GMPEs with associated weight factor were further used in probabilistic seismic hazard analysis of Patna SSA. These weight factors would further useful in forming the logic tree to reduce the epistemic uncertainty in final hazard value. Detailed analysis of determination of LLH and weight factor corresponding to each GMPE is given in Anbazhagan et al. (2015a).

## 5 Delineation and spatial smoothening of seismic source model

Various researchers have delineated the seismic source for various parts of India. Considering the tectonic features and the past earthquake events, Gupta (2006) delineated the seismic sources for India. Kiran et al. (2008) and NDMA (2010) have done the same on the basis on the seismicity parameters. Furthermore, Nath and Thingbaijam (2011) have delineated based on focal mechanism data from the Global Centroid Moment Tensor database. Vipin and Sitharam (2013) determined the seismic sources in peninsular considering the seismicity parameters. In the present study, delineation of the seismic sources

has been done based on the seismicity parameters viz. '$a$', '$b$' and magnitude of completeness ($M_c$). For delineation of different zones, Patna SSA has been divided into a grid size of 0.02°×0.02° and from the centre of each grid a radius of 500 km is considered. The number of earthquakes events within 500 km of each radius were considered to determine the seismicity parameters. The reason for selection of 500 km radius was discussed above and given in detail in Anbazhagan et al. (2015a, 2013a). Considering the seismicity parameters ($a$-value, $b$-value and $M_c$), the whole study area has been divided

into 7 areal seismic zones and shown in Figure 3 (variation of only $b$-value is shown in background). These seven zones are considered as areal seismic sources as these are spread over a large area. The seismicity parameter has been calculated for each of these zones considering the frequency magnitude distribution (FMD) at 90% confidence level. $M_{max}$ for each seismic zone has been calculated as per method discussed earlier. The average values of '$a$', '$b$', $M_c$ and $M_{max}$ have been given in Table 4.

For spatially smoothening of seismic source model, a grid size of 0.02°×0.02° along the longitude and latitude respectively was selected for representing different kinds of seismic source and to count number of earthquake with magnitude less than or equal to $M_c$ for each grid. To account the seismicity of the Patna SSA, the maximum likelihood estimates of $10^a$ for that grid cell has been determined which correspond to the number of earthquakes per year. Using maximum likelihood estimate of $10^a$, the recurrence rate for different magnitude intervals has been estimated using algorithm recommended by McGuire

and Arabasz (1990). The value $10^a$ for each grid has been smoothed by applying a Gaussian function, given as equation (1), to find the final modified values corresponding to each grid. This smoothing is made to account for the uncertainty related to the location of earthquake events.

$$ñ_i = \frac{\sum_j n_j e^{-\Delta_{ij}^2/c^2}}{\sum_j e^{-\Delta_{ij}^2/c^2}} \tag{1}$$

where, $n_j$ is the number of earthquake in the $j^{th}$ grid, $ñ_i$ is the smoothed number of earthquake in $i^{th}$ cell, c is the correlation

distance to account for the location uncertainties and $\Delta ij$ is the distance between the $i^{th}$ and the $j^{th}$ cell. The sum is taken over the $j^{th}$ cell should be within the distance of 3c of the $i^{th}$ cell.



## 6 Computation Models for determining hazard value

Probability of exceedance of a ground motion for a spectral period can be determined once the probability of its size, locations and level of ground shaking is identified cumulatively. Seismic hazard map for Patna district has been developed by applying probabilistic method namely classical method proposed by Cornell (1968) which was later improved by

5   Algermissen et al. (1982) and smoothed-gridded seismicity models (Frankel, 1995).

178 seismic sources (shown in Figure 1 and given as Table ET1) have been used for determining the probability of occurrence of a specific magnitude, probability of hypocentral distance and probability of ground motion exceeding a specific value have been estimatedas per Cornell (1968). Probability of rupture to occur at different hypocentral distances has been determined as per Kiureghian and Ang (1977). The condition probability of exceedence for GMPEs was

determined using a lognormal distribution as given by EM-1110 (1999). The ground motion at a site for a known probability of exceedence in a desired period has been calculated by amalgamating all the above probabilities. As a result of PSHA, hazard curve showing PGA or SA versus the frequency of exceedence of the level of ground motion. Detailed explanation is given in Anbazhagan et al. (2015 a). The deaggregation based on the principle of superposition has been proposed by Iyenger and Ghosh (2004) has been used. The probability of exceedence of ground motion for each seismic source has been

computed by merging these uncertainties. Detailed discussion on the methodology of PSHA can be found in Anbazhagan et al. (2009).

It can be noted that in the SSA, North-west and central part of Patna is not fully covered by well identified seismic sources and many sources given in the Figure 1 are not well studied to prove its seismic activity. Moreover, there are many places where linear source has not been identified. So, to overcome the limitation, zoneless approach proposed by Frankel (1995)

has been used for developing the PSHA map for Patna SSA. This method accounts the spatial smoothing of historic seismicity to directly calculate the probabilistic hazard. The annual rate of exceedence for a given ground acceleration level is given by equation 2

$$\lambda(Z > z) = \sum_d \sum_i 10^{[log_{10}(N_d/T) - b(m_i - m_{cut})]} P(Z > z/D_d M_i) \tag{2}$$

where, $d$ and $i$ are indices for distance and magnitude bins. $N_d$ is the total of ñ$_i$ values over a given hypocentral distance

increment (calculated using equation 1), $P(Z > z/D_d M_i)$ will give the probability that a PGA of $Z$ of will exceed $z$, when an earthquake of magnitude $M_i$ occur at a distance of $D_d$, $T$ is the time in years of earthquake catalogue used to determine $N_d$. The probability that a PGA of $Z$ of will exceed $z$ can be determined using by EM-1110, 1999. The hazard map has been determined by the four models proposed by Frankel, 1995. Model 1, Model 2 and Model 3 used for magnitude less than 7, however model 4 can be used for magnitude greater than 7. In model 1, the earthquake events having $M_w$ between 3 and 5

are assumed to illuminate areas of faulting which can produce destructive events. Model 2 also ensures that the hazard map reflects the local, historic rate of magnitude moment of 5 and larger events. As this model cannot explain the cause of major earthquake in the Active region with certainty, it is prudent to address the possibility of near-repeats i.e. within about 100 km of an historic moderate earthquake. Model 3 is based on a uniform source zone encompassing the Active seismicity zone,




which is opposite to model 2. Model 4 associated with hazard from the larger events that is $M_w > 7$. As these events are less in the active seismic region and limited to a few areas only, therefore sources associated with them has been considered for determining hazard. These models are shown in Figure 4 which is used for the development of PSHA map using method proposed by Frankel (1995).

**7 Modelling of Logic tree for hazard analysis**

Seismic hazard can be assessed more practically using logic tree (Kulkarni et al., 1984) as it includes the accounted epistemic errors, components of seismic models and ground motion predictions (Figure 5). For determining the consistent model with different degrees of confidence each branch of logic tree is to be investigated for implementing the uncertainties in probability models. The important consideration has been given to each branch of logic tree by incorporating the respected

weights for assessing the final hazard of Patna district. After declustering the catalogue and developing the seismotectonic map, two models have been used with an equal weight of 50% for both classical and zone less approach. Zone less approach has been further divided as areal approach and Frankel approach of equal weight of 50% each. For Frankel approach, SSA has been considered for four models (discussed above) with weight factor of 30%, 30%, 20% and 20% for model 1, model 2, model 3 and model 4 respectively. These weights have been adopted based on the reliability of the source model. Larger

weights are assigned to model 1 and model 2 because they are based on more reliable data and assumedly better representation of seismicity of SSA. Model 3 deals with the weak assumption that earthquakes with magnitude 3.0-7.0 are equally probable everywhere in Patna SSA whereas there is a great uncertainty in the data used for model 4. In addition, b-value were calculated for each of the model using Gutenberg and Richter (1956) and Woessner and Wiemer, (2005) (using entire magnitude range method) by assigning equal weight factor of 0.5. Furthermore $M_{max}$ has been calculated using three

methods namely increment to $M_{max}^{obs}$, Kijko and Sellevoll (1989) and regional rupture characteristic with weight factor of 30%, 30% and 40% respectively for each model as shown in Figure 5. Segmented based analysis of GMPE was done and weight was assigned to each GMPE based on the efficacy test. Based on the above discussion final hazard map for Patna SSA has been produced for 2% and 10% probability of exceedance in 50 years.

**8 Mapping of probability of exceedence using different approach considering epistemic uncertainty**

**8.1 Classical Approach (Cornell, 1968)**

For determining the hazard value, different weight has been considered with respect to b-value, maximum magnitude and GMPE (see Figure 5). The seismic hazard using classical approach (Cornell, 1968) has been estimated using 178 seismic sources. SSA is divided into1725 grids of size 0.02°×0.02°. The whole procedure can be referred from Anbazhagan et al. (2015 a). Hazard curve from 10 most venerable sources are given as Figure 6 (a) and S60 is determined as most venerable

for Patna district (7.5 $M_w$ and hypocentral distance 55.11 km). Figure 6b showed a cumulative hazard curve obtained at the



Patna district centre for zero s, 0.05 s, 0.1 s, 0.2 s, 0.3 s, 0.4 s, 0.6 s, 0.8 s, 1.0 s, 1.6 s and 2 s. It can be observed from the Figure 6b, that the frequency of exceedance for 0.075 g at zero second is 0.001 which will give the return period 834 years. This indicates that PGA of 0.075 g has 5.03% probability of exceedence in 50 years at the Patna. Further explanation can be referred from Anbazhagan et al. (2015 a). The mean deaggregation plot for Patna for return period of 2475 and 475 years is

given as Figure 7a and 7b. PGA for 6.0 $M_w$ at 40 km hypocentral distance is notable for 2% probability of exceedence at 50 years. Likewise, for 10% probability of exceedence at 50 years the motion for 5.5 $M_w$ at 50 km hypocentral distance is most contributing. Hazard curve has been generated at each grid for Patna, and the level of ground motion for frequency of exceedence'$\nu(z)$' can be estimated from it. Figure 8a and 8b shows the PSHA maps for Patna district for return period of 2475 and 475 years respectively. PGA varies from 0.35g in the north western and 0.43 north eastern peripheries to 0.08g

towards the central part (See Figure 8a). Similarly, PGA vale at north eastern periphery is 5.3 times more than central part of Patna considering 10% probability in 50 years (see Figure 8b). These results are similar to the previous study done by Anbazhagan et al. (2015 a).

**8.2 Zoneless Approach**

Likewise, classical approaches, epistemic uncertainty has been considered and weight factor are considered as shown in

Figure 5. The PGA map of Patna has been developed using zoneless approach by dividing it into seven areal zones based on seismicity-parameters (Figure 3). For the development of PSHA map using simplified areal zonal modal, the seven zones along with the seismic parameters (Figure 3 and Table 4) are used. These seven areal seismic sources are smoothed using smoothed historic seismicity approach recommended by Frankel (1995). For development of the seismic hazard map, grid size of 0.02°×0.02° was selected for each of these seven areal sources. The activity rate was calculated in every grid cell and

it was obtained by counting the earthquake having magnitude greater than or equal to $M_c$ (Table 4) for the whole earthquake catalogue using MATLAB. The calculated activity rate was then spatially smoothed according to Equation 1, and the chosen correlation distance $c = 50$ km. The annual rate of exceedence at the centre of each grid for the seven zones has been calculated using equation 2. The cumulative hazard curves for different period at the Patna district centre is given as Figure 9. At zero period, frequency of exceedence for 0.075 g is 0.012 and estimated return period is 84 years, which means 0.075 g

has 44.96 % probability of exceedence in 50 years. Similarly, for 0.5 g, return period is 24.4 thousand years and probability of exceedence of 2.05 x $10^{-1}$ % in 50 years at Patna district centre. As the period on interest rises from zero second to 0.8 seconds, a huge change in return period has been noticed (see Figure 9). Primarily the frequency of exceedence decreases from 84 years at zero periods to 13 years at 1.0 second which has further increased to 28 years at 0.2 second and again till 1.97E+05 years for 2 second. The mean deaggregation plot for Patna SSA for return period of 2745 and 475 years is given as

Figures 10a and 10b. Figure 10a shows that the motion for 6.0 $M_w$ at 15 km hypocentral distance is dominant for 2% probability of exceedence at 50 years. It changed to 5.75 $M_w$ at 20 km hypocentral distance for 10% probability of exceedence at 50 years. Figures 11a and 11b are the PSHA maps for Patna urban centre for 2 % and 10 % probabilities of exceedence in 50 years respectively considering zoneless approach. PGA varies from 0.41 g in the south-eastern periphery to





0.34 g towards the central part (See Figure 11a). However, southwest part of the district encounters PGA of 1.4 times that of northwest part of the district. Similar PGA at southwest part increases to 1.57 folds as compared to north western part while considering 10% probability of exceedence in 50 years (Figure 11b).

**8.3 Four models (Figure 4) using Zoneless Approach (Frankel, 1995)**

The hazard value for Patna district has also been determined by the four-model proposed by Frankel (1995). Each of these four models (Figure 4) has different spatial distribution of seismic activity. However present SSA have 5 characteristic earthquakes ($M_w \geq 7$) so model 1, 2 and 3 have been analysed separately by considering earthquake events and PGA map using model 4 have been developed based on seismic sources associated with characteristic earthquake. The seismic hazard map is generated considering grid size of 0.02°×0.02°. The activity rate was calculated in every grid cell and it has been

obtained by counting the earthquake having magnitude greater than or equal to $M_c = 3.0$ & 5.0 for Model 1 and Model 2 & 3 for different period of earthquake catalogue (Figure 4) using MATLAB. The calculated activity rate was then spatially smoothed according to Equation 1, and the chosen correlation distance $c = 50$, 75 km for model 1 and model 2 & 3. The annual rate of exceedence at the centre of each grid for the seven zones has been calculated using equation 2. The cumulative hazard curve has been obtained from model 1, 2, 3 and 4 at the Patna district centre for zero s, 0.05 s, 0.1 s, 0.2 s, 0.3 s, 0.4 s,

0.6 s, 0.8 s, 1.0 s, 1.6 s and 2 s and shown in Figure 12. At zero period, return period is 85 years and 0.075 g have 43.96 % probability of exceedence in 50 years at the Patna district centre and 0.5 g, return period increased 24.4 thousand years. Primarily the frequency of exceedence declines from 85 years at zero periods to 14 years at 1.0 seconds which has further increased to 29 years at 0.2 seconds and again till 2.0E+05 years for 2 second. Figures 13a and 13b shows the mean deaggregation plot for Patna for 2% and 10% probability of exceedence at 50 years. PGA for 6.0 $M_w$ at 25.25 km

hypocentral distance and 5.75 $M_w$ at 30.3 km hypocentral distance is predominant for 2 and 10% probability of exceedence at 50 years. With the four models described in Figure 4, PGA map has been developed for Patna SSA and given in Figure 14a, 14b, 14c& 14d considering 2% probability of exceedence in 50 years and Figure 15a, 15b, 15c &15d considering 10% probability of exceedence in 50 years. It can be noted from model 1 that south-western part of Patna has high hazard value similar trend has been seen from model 2. The model 3 is a map of uniform hazard whereas as far as model 4 is concerned,

north-eastern part and central part have high hazard because that portion of SSA is associated with characteristic earthquakes. The weighted mean PGA map for Patna has been developed by assigning different weight to these 4 models as 0.3, 0.3, 0.2 and 0.2 for model 1, 2, 3 and 4 respectively. A larger weight is given to model 1 and 2 as they represent real seismic activity because they are based on more reliable data. However, model 3 deals with weak conjecture that earthquake events between 3 to 7 are equally likely everywhere in Patna and Model 4 has great uncertainty in occurrence of

characteristic earthquake. Figures 16a and 16b are the PSHA maps for Patna district for return period of 2475 and 475 years respectively. PGA varies from 0.34g in the eastern periphery to 0.26 g towards the north-western periphery, while increases to 1.38-fold for southwest part of the district (see Figure 16a). Similarly, considering 10% probability of exceedence in 50 years, PGA value in south western part of Patna is 1.5 times the south-western part (see Figure 16b).





It has seen from the mean deaggregation plot that the motion for 6.0 $M_w$ at 40 km hypocentral distance, 6.0 $M_w$ at 15 km hypocentral distance and 6.0 $M_w$ at 25.25 km hypocentral distance is predominant in case of Cornel's, Areal and Frankel's approach respectively considering 2 % probability in 50 years. However, the motion for 5.5 $M_w$ at 50 km hypocentral distance, 5.75 $M_w$ at 20 km hypocentral distance and 5.75 $M_w$ at 30.3 km hypocentral distance respectively predominant in
case of Cornel's, Areal and Frankel's approach. The PGA values varies from 0.08 to 0.43 g, 0.29 to 0.41 g and 0.26 to 0.36 g in case of Cornel's, Areal and Frankel's approach respectively considering 2 % probability in 50 years. Whereas it from 0.04 g to 0.18 g, 0.09 g to 0.16 g and 0.09 g to 0.16 g respectively considering 10 % probability of exceedence in 50 years in case of Cornel's, Areal and Frankel's approach. On comparing hazard map developed using classical approach and zoneless approach, it has been seen that north-eastern part of Patna SSA has experienced maximum PGA value. As per classical
approach (Cornell, 1968), predicted PGA value for central part of Patna district is 0.08 g whereas per Frankel's approach (Frankel, 1995) approach it is 0.32 g, however as per areal approach it is 0.31 g. Similarly, PGA value of 0.15g, 0.39 g and 0.39 g has been observed in case of Cornel's, Frankel's and Areal approach approximately in south western part of Patna SSA. It is because of absence of well-defined seismic source in that area whereas earthquake events of moment magnitude of 6 and above have occurred. However, in north western part PGA value is almost equal calculating using these approaches.
This is the reason both zoneless and classical approach has been considered in this study to counter the epistemic uncertainty. So, that both the seismic sources and earthquake events can be accounted properly.

## 9 Final hazard map using Logic tree approach

The final hazard value has been developed by assigning the weight factor or 0.5 to both PGA value calculated corresponding to classical and zoneless approach. In zoneless approach, 0.5 weight factor were given to both PGA map developed using
areal and Frankel's (1995) approach as explained earlier. So, both the hazard maps were compiled and finally 0.5 weight factor is given to zoneless approach. The final PGA variation corresponds to 2% and 10% probability of exceedance in 50 years were shown as Figures 17a and 17b. In addition to that SA at respectively 0.2 and 1 s considering epistemic uncertainty has been given as Figure 18a, 18b, 18c, and 18d for 2% and 10% probability of exceedence in 50 years respectively. PGA varies from 0.37 g in the south-eastern periphery to 0.30 g towards the northwest periphery, whereas
southwest part of the district encounters PGA of 0.31 g (See Figure 17 a). Similarly, PGA corresponding to 475 years return period is about 0.12 g in the north-western periphery and 0.15 g in the south-eastern periphery (Figure 17b). The reason for having maximum PGA value in the south-eastern periphery is due to the location of East Patna and West Patna Fault and PGA value of 0.35 g in south western part is due to the presence of earthquake events of magnitude moment more than 6. PGA value varies from 0.12 to 0.15 g for a return period of 2475 year which is comparable with PSHA map of India
developed by Nath and Thingbaijam (2012). Recently, a major thrust faulting earthquake of magnitude 7.8 on 25 April 2015 occurred in Nepal which affected various place in India including Patna district is one of them. We have completed our



mapping before this earthquake and compared our results with shake map published by USGS (2015). It is noticed that PGA values for 10% probability of exceedence in 50 years is matches with USGS (2015) shake map on recent Nepal Earthquake. In addition to that, uniform hazard response spectrum (UHRS) has been developed considering all the three approaches and compared with IS 1893 (2002). For developing UHRS, seismic hazard curves of spectral accelerations at different spectral period for the same probability of exceedence has been developed. The UHRS at 2 and 10 % probability of exceedence for 50 years at the centre of the district using classical and zoneless approach viz. Frankel's and areal approach has been drawn and given as figure 19 a (marked as star in figure 17 a). Similarly, UHRS has been developed at the North-eastern part of Patna considering 2 and 10 % probability of exceedance, shown as figure 19 b (marked as plus in figure 17 a). It has been seen from Figure 19 that the hazard value at 2 % probability is more for the same return period when compared to 10% probability of exceedence in 50 years. It has been also observed that spectral acceleration at zero period i.e. PGA is less in case of Cornell's approach when compared to Frankel's and Areal approach at the centre of the district where as it is more when compared to the North-eastern part of SSA. The developed UHRS has been compared with IS 1893 (2002) and it has been observed that the SA predicted is lower at the centre of the district at 2 and 10 % probability of exceedence in 50 years except for Frankel's approach. However, in case of North eastern parts of SSA, the predicted SA values are more as compared to IS 1893 (2002) (Figure 19 b). Hence, UHRS should be developed based on the regional characteristics so that it could be effectively used in infrastructural development of a district.

## 10 Conclusion

A new seismic hazard map for Patna district was developed considering the earthquake events and seismic sources through logic tree approach. Based on past earthquake damage distribution, seismic study area of 500 km was arrived and the seismotectonic map was generated. The maximum magnitude has been estimated by considering weighted mean three methods, i.e. incremental method, Kijko method and regional rupture-based characteristic. From 27 applicable GMPEs, GMPEs ANBU-13, NDMA-10 and KANO-06 were selected upto 100 km epicentral distance, however ANBU-13, NDMA-10, BOAT-10 and KANO-06 up to 300 km and NDMA-10 for more than 300 km. These GMPEs were ranked and weights were found based on the Log-Likelihood method. A new hazard map for Patna district has been developed using both classical and zoneless approach considering different weight factor corresponds to b-value, maximum magnitude and GMPE to reduce the uncertainty values. The logic tree has been accounted to capture this epistemic uncertainty in the seismicity models. The final seismic hazard map corresponding to 2% and 10% probability of exceedence in 50 years has been developed by giving weight factor to the seismicity models, maximum magnitude and GMPEs. The PGA values varies from 0.08 to 0.43 g, 0.29 to 0.41 g and 0.26 to 0.36 g in case of Cornel's, Areal and Frankel's approach respectively considering 2 % probability in 50 years. Whereas it from 0.04 g to 0.18 g, 0.09 g to 0.16 g and 0.09 g to 0.16 g respectively considering 10 % probability of exceedence in 50 years in case of Cornel's, Areal and Frankel's approach. However, hazard values in terms of PGA at bed rock level after considering logic tree varies from 0.30 to 0.37 g and 0.11 to 0.15 g respectively considering 2



and 10 % probability of exceedence in 50 years. In addition to that spectral acceleration hazard map has been developed at a period of 0.2 and 1 s corresponds to 2% and 10 % probability of exceedence in 50 years. Hence the logic tree should be used to reduce the epistemic uncertainty in determining the hazard value for any seismic study area. It has been also concluded that uniform hazard response spectra should be developed considering regional specific parameters.

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

**List of Figures**





**Figure 14: PGA map of Patna SSA considering 2% probability in 50 years for model 1 (a), model 2 (b), model 3 (c) and model 4 (d) using Frankel approach (1995)**

**Figure 15: PGA map of Patna SSA considering 10% probability in 50 years for model 1 (a), model 2 (b), model 3 (c) and model 4 (d) using Frankel approach (1995)**

**Figure16 (a): Weighted mean PGA map of Patna SSA for 2% probability of exceedence in 50 years using Frankel approach (1995)**

**Figure 16 (b): Weighted mean PGA map of Patna SSA for 10% probability of exceedence in 50 years using Frankel approach (1995)**

**Figure 17 (a): Final seismic hazard map of Patna SSA for 2% probability of exceedence in 50 years using Logic tree approach**

**Figure 17 (b): Final seismic hazard map of Patna SSA for 10% probability of exceedence in 50 years using Logic tree approach**

**Figure 18: Final seismic hazard map of Patna SSA for (a) 2% and (b) 10% probability of exceedence in 50 years at 0.2 s respectively and (c) 2% and (d) 10% probability of exceedence in 50 years at 1 s respectively using Logic tree approach**

**Figure 19 (a): Design spectrum for Patna for 5% damping from 2% and 10% probability of exceedence in 50 years and IS 1893 (2002) at centre of the city (marked in Figure 17 a)**

**Figure 19 (b): Design spectrum for Patna for 5% damping from 2% and 10% probability of exceedence in 50 years and IS 1893**
**(2002) at north eastern part of the city (marked in Figure 17 a)**

**List of Tables**

**Table 1: Variation in Magnitude of Completeness ($M_c$), a and b parameter of G-R Relationship**

**Table 2: Available GMPEs with their Abbreviations considered for the seismic study area**

**Table 3: Segmented Ranking of GMPEs for Patna Region**

**Table 4: Seismic parameters for adopted source models (uncertainties with bootstrapping)**

**List of Electronic Material**

**Table S1: $M_{max}$ corresponds to seismic sources used in the hazard analysis**



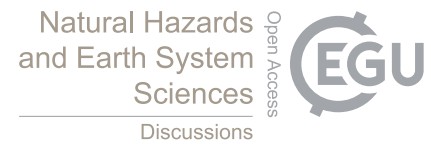
Table 1 Variation in Magnitude of Completeness ($M_c$), a and b parameter of G-R Relationship

**Gutenberg and Richter (1956)**

| | $a-value$ | | $b-value$ | |
|---|---|---|---|---|
| | Region I | Region II | Region I | Region II |
| | 5.32 | 4.98 | 0.87 | 0.97 |

**Woessner and Wiemer (2005)**

| S. No. | Methods | $M_c$ | | $a-value$ | | $b-value$ | |
|---|---|---|---|---|---|---|---|
| | | Region I | Region II | Region I | Region II | Region I | Region II |
| 1 | M1 | 4.7 | 4.7 | 6.28 | 6.22 | 0.789 | 0.815 |
| 2 | M2 | 1.7 | 1.9 | 3.11 | 3.07 | 0.149 | 0.176 |
| 3 | M3 | 4.6 | 4.7 | 5.92 | 6.22 | 0.720 | 0.815 |
| 4 | M4 | 4.9 | - | 6.50 | - | 0.833 | - |
| 5 | M5 | 5.0 | 4.7 | 6.57 | 6.22 | 0.843 | 0.815 |
| 6 | M6 | 5.0 | 4.8 | 6.57 | 6.21 | 0.843 | 0.814 |
| 7 | M7 | 1.9 | 4.7 | 3.41 | 6.22 | 0.214 | 0.815 |
| 8 | M8 | 1.8 | 4.9 | 3.13 | 6.40 | 0.154 | 0.848 |
| 9 | M9 | 1.8 | 2.0 | 3.13 | 3.13 | 0.154 | 0.190 |



Table 2: Available GMPEs with their Abbreviations considered for the seismic study area

| S. No. | Ground Motion Prediction Equation (GMPE) | Abbreviation of the equations |
|---|---|---|
| 1. | Singh et al. (1996) | SI-96 |
| 2. | Sharma (1998) | SH-98 |
| 3. | Nath et al. (2005) | NATH-05 |
| 4. | Das et al. (2006) | DAS-06 |
| 5. | Sharma and Bungum (2006) | SHBU-06 |
| 6. | Baruah et al. (2009) | BA-09 |
| 7. | Nath et al. (2009) | NATH-09 |
| 8. | Sharma et al. (2009) | SH-09 |
| 9. | Gupta (2010) | GT-10 |
| 10. | National Disaster Management Authority, (2010) | NDMA-10 |
| 11. | Anbazhagan et al. (2013b) | ANBU-13 |
| 12. | Abrahamson and Litehiser, (1989) | ABLI-89 |
| 13. | Youngs et al. (1997) | YONG-97 |
| 14. | Campbell (1997) | CAMP-97 |
| 15. | Spudich et al. (1999) | SPUD-99 |
| 16. | Atkinson and Boore (2003) | ATKB-03 |
| 17. | Takahashi et. al. (2004) | TAKA-04 |
| 18. | Ambraseys et al. (2005) | AMB-05 |
| 19. | Kanno et al. (2006) | KANO-06 |
| 20. | Zhao et al. (2006) | ZHAO-06 |
| 21. | Campbell and Bozorgnia (2008) | CABO-08 |
| 22. | Idriss (2008) | IDRS-08 |
| 23. | Boore and Atkinson (2008) | BOAT-08 |
| 24. | Abrahamson and Silva (2007) | ABSI-08 |
| 25. | Aghabarati and Tehranizadeh (2009) | AGTE-08-09 |
| 26. | Lin and Lee (2008) | LILE-08 |
| 27. | Akkar and Bommer (2010) | AKBO-10 |




Natural Hazards and Earth System Sciences Discussions — Open Access

Table 3 Segmented Ranking of GMPEs for Patna Region

| Sl. No. | GMPEs | 0-100 1833 LLH | 0-100 1833 DSI | 0-100 1833 Ranking | 0-100 1934 LLH | 0-100 1934 DSI | 0-100 1934 Ranking | 100-300 1833 LLH | 100-300 1833 DSI | 100-300 1833 Ranking | 100-300 1934 LLH | 100-300 1934 DSI | 100-300 1934 Ranking | 300-500 1833 LLH | 300-500 1833 DSI | 300-500 1833 Ranking |
|---|---|---|---|---|---|---|---|---|---|---|---|---|---|---|---|---|
| 1 | ABLI-89_Hort | 16.68 | -99.95 | 15 | 28.33 | -100 | 11 | | NA | | | NA | | | NA | |
| 2 | ABLI-89_Vert | 17.77 | -99.98 | 16 | 26.38 | -100 | 10 | | NA | | | NA | | | NA | |
| 3 | YONG-97 | 26.16 | -100 | 22 | 53.45 | -100 | 15 | 8.06 | -64.67 | 5 | 17.87 | -99.75 | 6 | 6.50 | -76.14 | 2 |
| 4 | CAMP-97 | 8.47 | -85.22 | 12 | 13.28 | -97.79 | 5 | | NA | | | NA | | | NA | |
| 5 | SPUD-99 | 19.11 | -99.99 | 17 | | NA | | | NA | | | NA | | | NA | |
| 6 | TAKA-04 | 9.95 | -94.71 | 10 | 15.91 | -99.64 | 7 | 11.16 | -95.89 | 8 | 15.87 | -99.01 | 6 | | NA | |
| 7 | AMB-05 | 22.96 | -100.00 | 20 | 70.48 | -100 | 16 | | NA | | | NA | | | NA | |
| 8 | NATH-05 | 20.71 | -100.00 | 18 | 39.57 | -100 | 12 | | NA | | | NA | | | NA | |
| 9 | KANO-06 | **5.45** | **19.29** | **3** | 10.91 | -88.56 | 4 | **4.58** | **295.22** | **1** | **6.98** | **370.44** | **2** | | NA | |
| 10 | ZHAO-06 | 9.95 | -94.71 | 9 | 15.91 | -99.64 | 6 | 11.16 | -95.89 | 7 | 15.87 | -99.01 | 5 | | NA | |
| 11 | SHBU-06 | 5.75 | -2.93 | 4 | | NA | | 26.39 | -100 | 10 | | NA | | | NA | |
| 12 | IDRS-08 | 14.84 | -99.82 | 14 | 25.58 | -100 | 9 | 29.29 | -100 | 11 | 45.51 | -100 | 9 | | NA | |
| 13 | BOAT-08 | 11.13 | -97.66 | 13 | 19.34 | -99.97 | 8 | **5.85** | **63.24** | **4** | 10.00 | -41.68 | 4 | 10.32 | -98.31 | 4 |
| 14 | ABSI-08 | 9.31 | -91.78 | 8 | 44.92 | -100 | 14 | 16.12 | -99.87 | 9 | | NA | | | NA | |
| 15 | CABO-08 | 24.56 | -100.00 | 21 | 72.44 | -100 | 18 | 51.43 | -100 | 12 | 149.01 | -100 | 10 | | NA | |
| 16 | LILE-08 | 22.72 | -100.00 | 19 | 42.91 | -100 | 13 | 10.62 | -94.01 | 6 | 22.04 | -99.99 | 8 | 8.55 | -94.23 | 3 |
| 17 | AGTE-08-09_Vert | 7.69 | -74.76 | 7 | | NA | | | NA | | | NA | | | NA | |
| 18 | AGTE-08- | 6.95 | -57.77 | 6 | | NA | | | NA | | | NA | | | NA | |


|  | 09_Hort |  |  |  |  |  |  |  |  |  |  |  |  |  |  |  |
|---|---|---|---|---|---|---|---|---|---|---|---|---|---|---|---|---|
| 19 | NATH-09 | 5.79 | -5.24 | 5 | 9.87 | -76.50 | 3 | NA | NA |  | NA | NA |  | NA | NA |  |
| 20 | AKBO-10 | 10.15 | -95.41 | 11 | NA |  |  | NA | NA |  | NA | NA |  | NA | NA |  |
| 21 | NDMA-10 | 2.81 | 645.32 | 2 | 6.09 | 222.97 | 2 | 4.84 | 229.98 | 3 | 9.74 | -30.27 | 3 | 2.55 | 268.69 | 1 |
| 22 | ANBU-13 | 2.23 | 1011.90 | 1 | 4.04 | 1241.71 | 1 | 4.70 | 261.84 | 2 | 6.90 | 399.06 | 1 | NA |  |  |



Table 4 Seismic parameters for adopted source models (uncertainties with bootstrapping)

| Source Model | | | | |
|---|---|---|---|---|
| Zone N | a-value* | b-value* | $M_c$* | $M_{max}$ (Kijko and Sellevoll (1989) |
| 1 | 5.17 | 0.885 | 4.5 | 6.7 |
| 2 | 5.15 | 0.910 | 4.2 | 6.4 |
| 3 | 5.04 | 0.910 | 4.2 | 6.4 |
| 4 | 5.07 | 0.848 | 4.6 | 7.0 |
| 5 | 5.01 | 0.880 | 4.3 | 6.5 |
| 6 | 5.12 | 0.880 | 4.3 | 6.5 |
| 7 | 5.01 | 0.878 | 4.3 | 6.5 |

* Average value of seismicity parameters



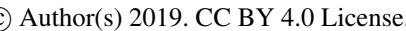



Figure 1: Seismotectonic Map of Patna SSA



Figure 2: $M_c$ value along with $'a'$ and $'b'$ parameters for M1, M2, M3, M4, M5, M6, M7, M8 and M9 methods for region I (open square) and region II (open triangle)





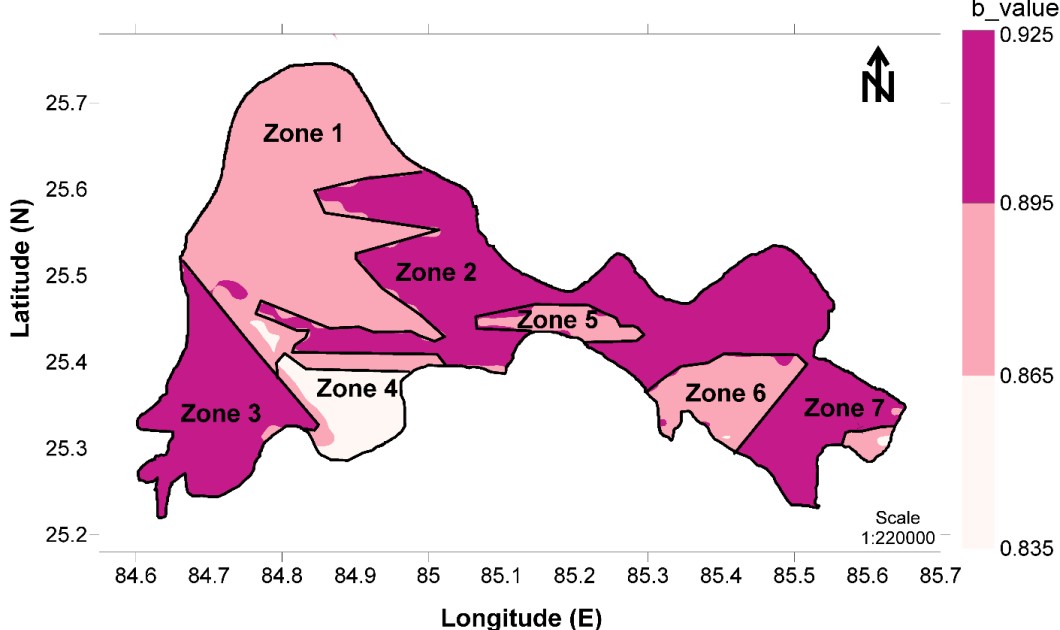

Figure 3: New seismic source zones identified for Patna based on seismicity parameters
(variation of 'b' value is shown in background)




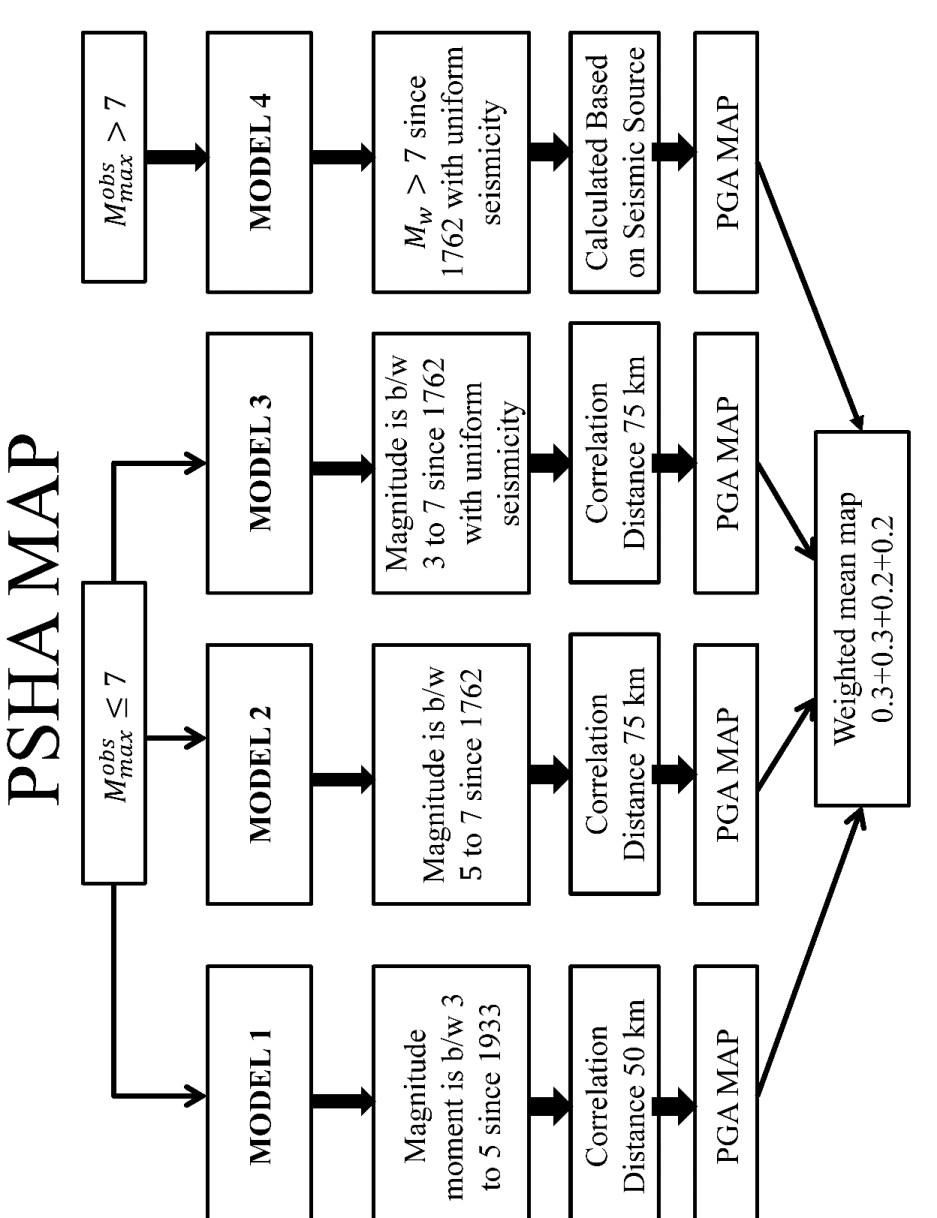

Figure 4: Four models used in the development of PSHA map of Patna based on Zoneless Approach





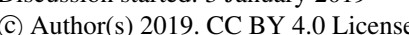

Figure 5: Formulated logic tree used in PSHA of Patna SSA





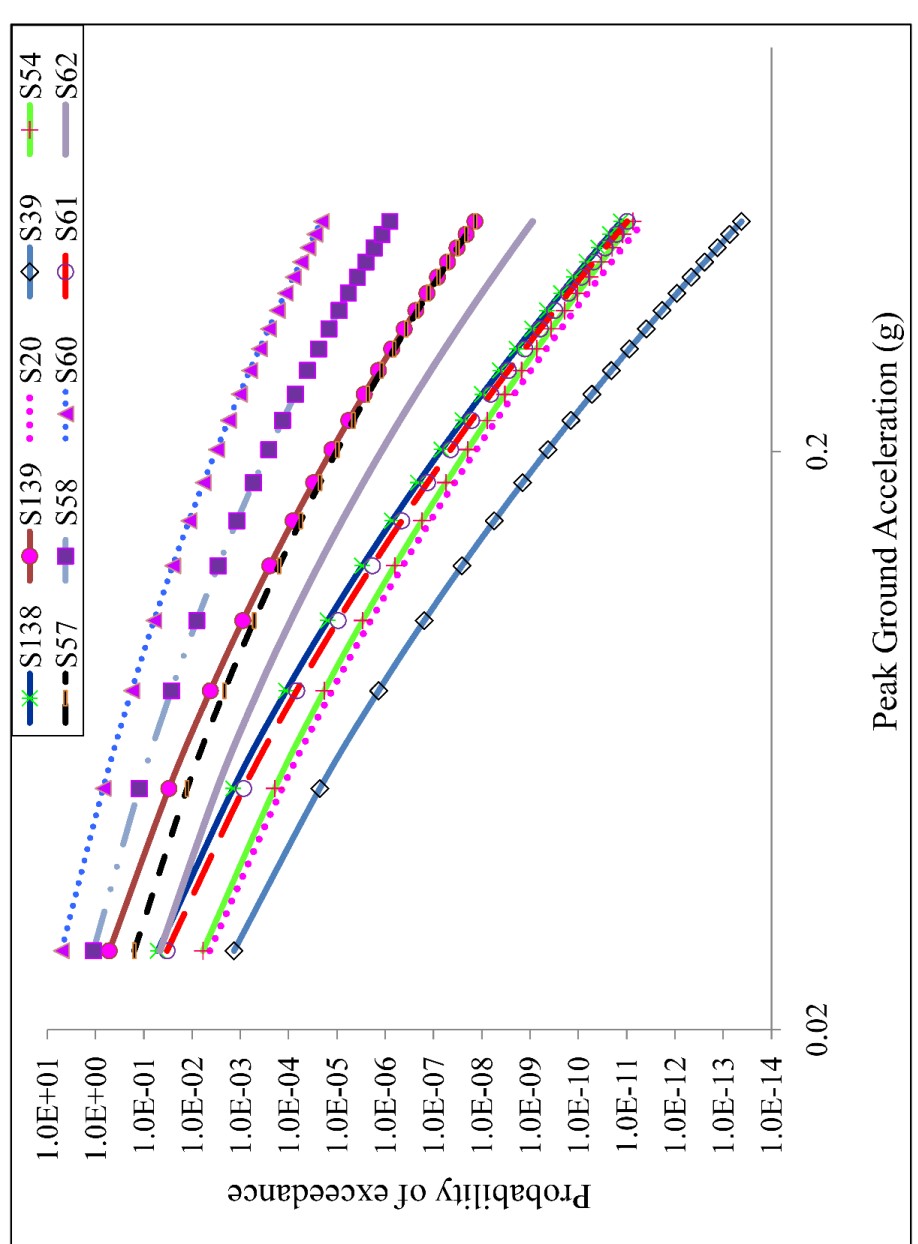

Figure 6 (a): Hazard Curve for ten most contributing seismic source at Patna city centre



Figure 6 (b): Hazard curve at Patna district centre for different periods using Classical Approach

6(b)

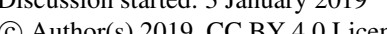



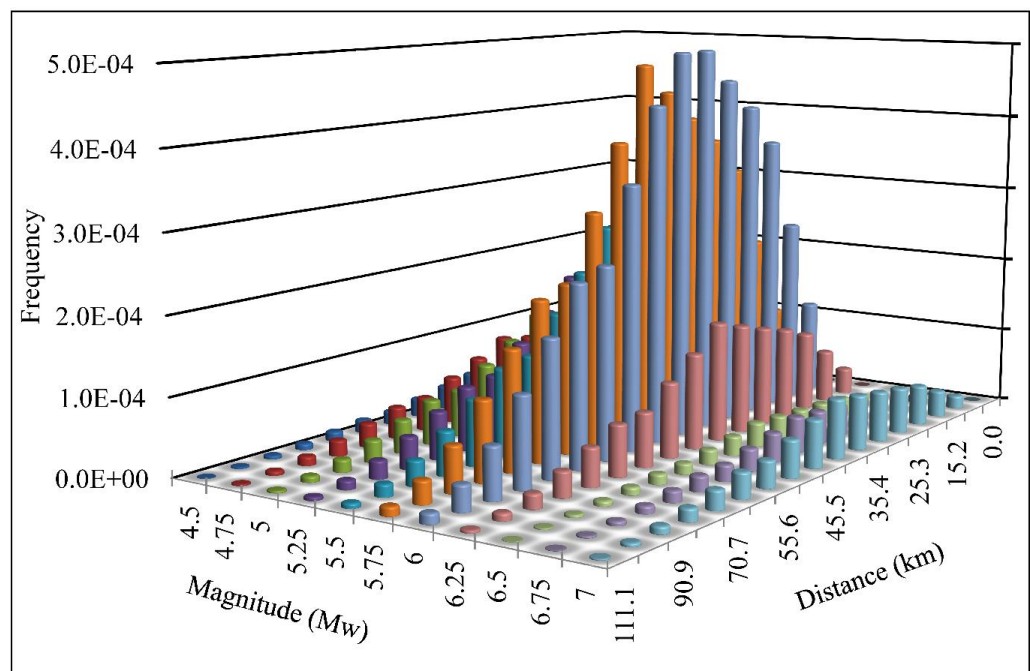

Figure 7 (a): Deaggregation of hazard value at Patna at bed rock at PGA for 2 % probability of exceedence in 50 years using Classical Approach




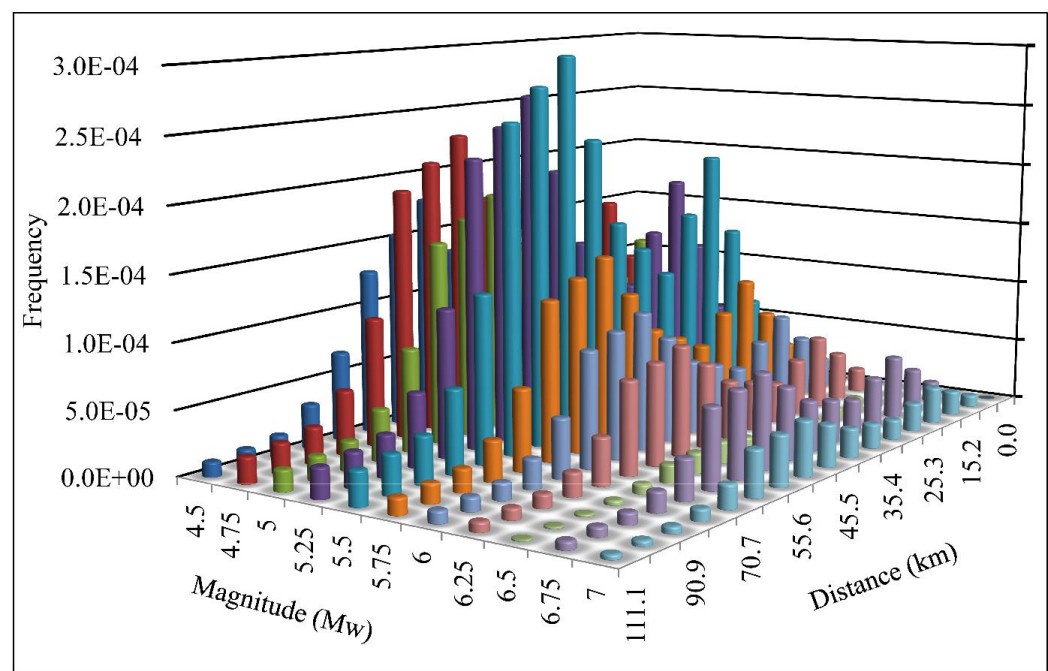

Figure 7 (b): Deaggregation of hazard value at Patna at bed rock at PGA for 10 % probability of
exceedence in 50 years Classical Approach



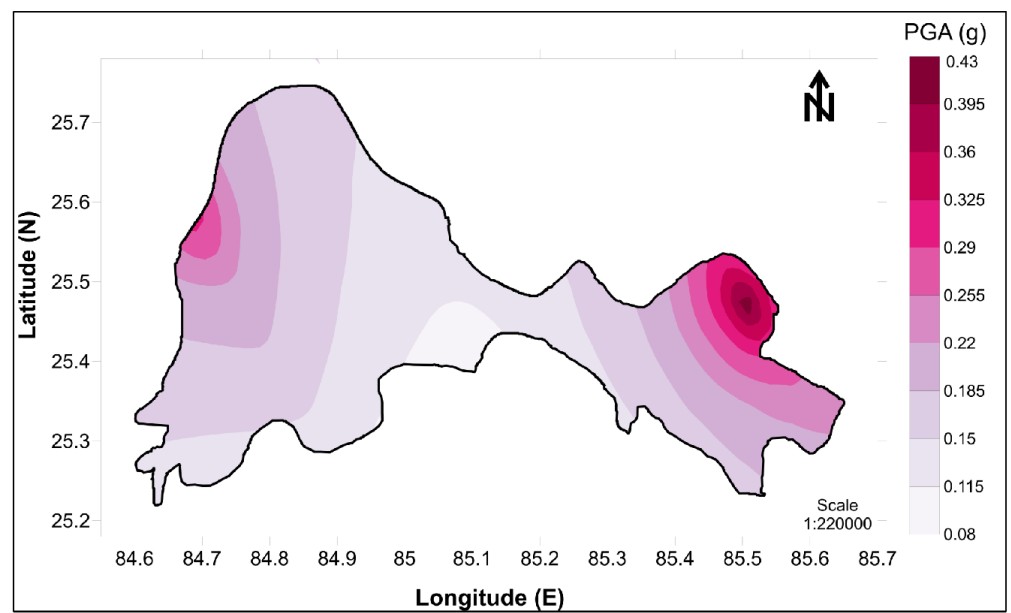

Figure 8 (a): PSHA map for Patna urban centre for 2 % probability of exceedence in 50 years using classical approach

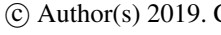



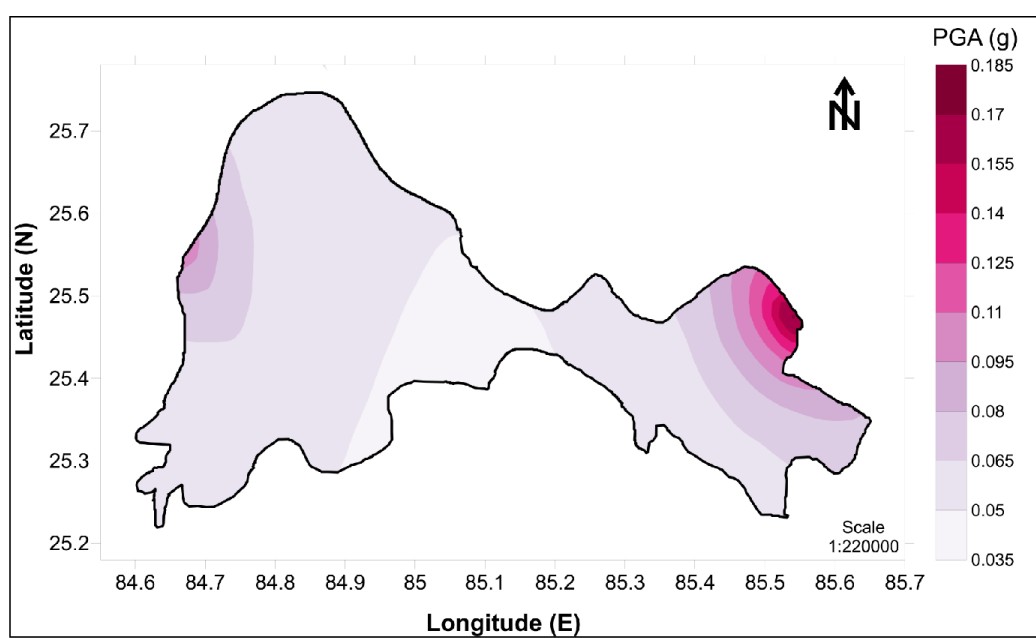

Figure 8 (b): PSHA map for Patna urban centre for 10 % probability of exceedence in 50 years
using classical approach







Figure 9: Hazard curve at Patna district for different periods using areal seismic zone (considering centre of zone 2)





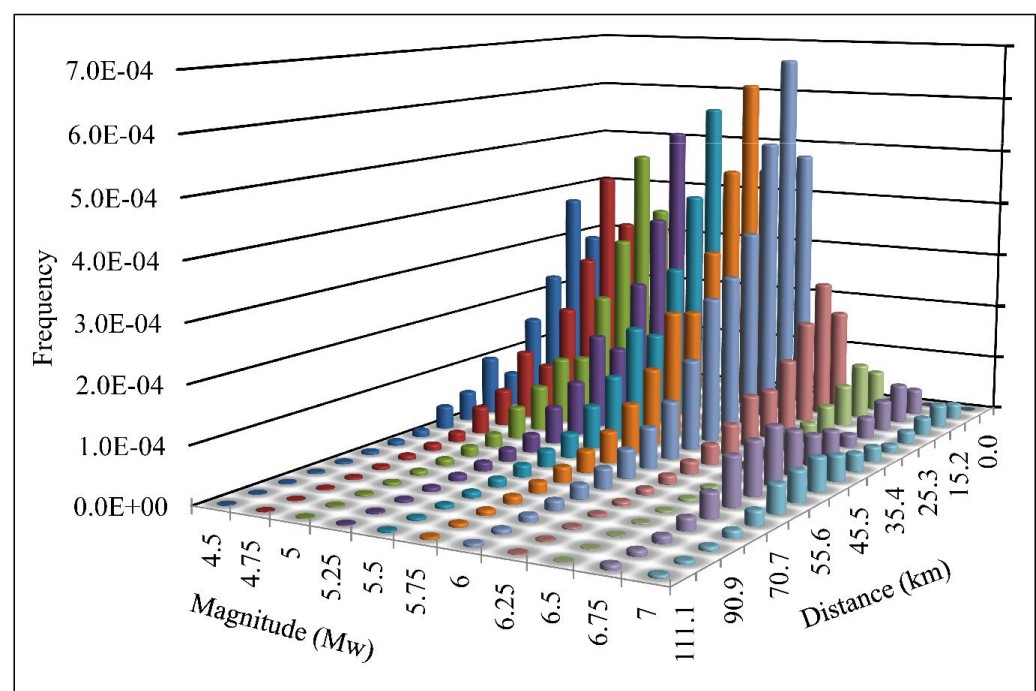

Figure 10 (a): Deaggregation of hazard value at Patna at bed rock at PGA for 2 % probability of exceedence in 50 years using areal seismic zone





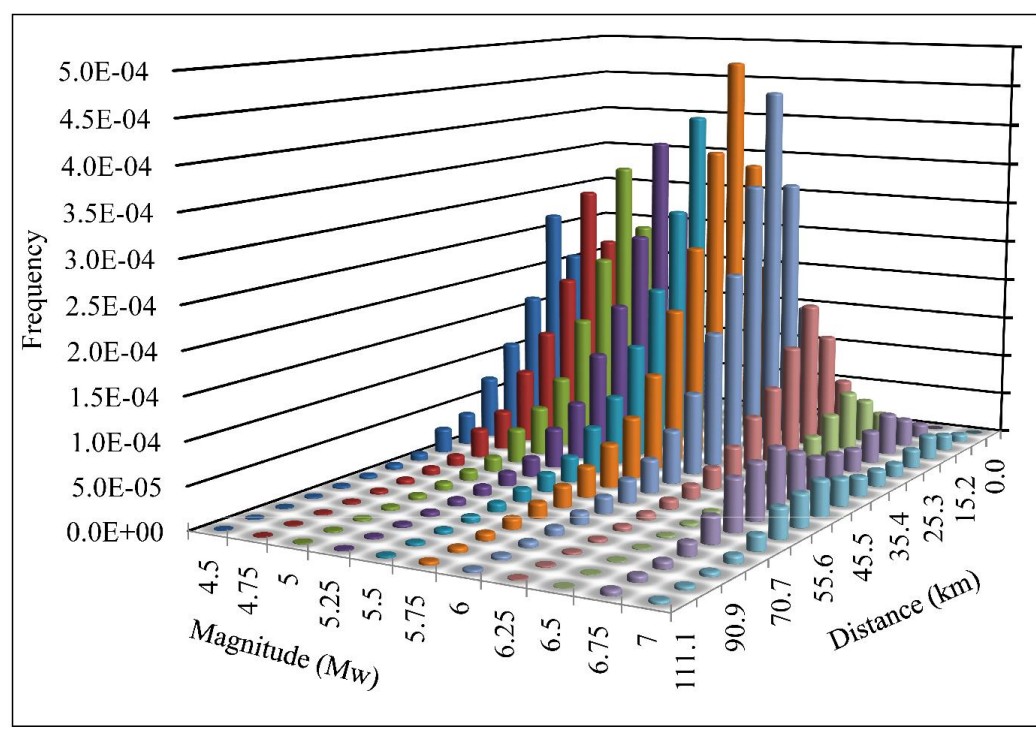

Figure 10 (b): Deaggregation of hazard value at Patna at bed rock at PGA for 10 % probability of exceedence in 50 years using areal seismic zone




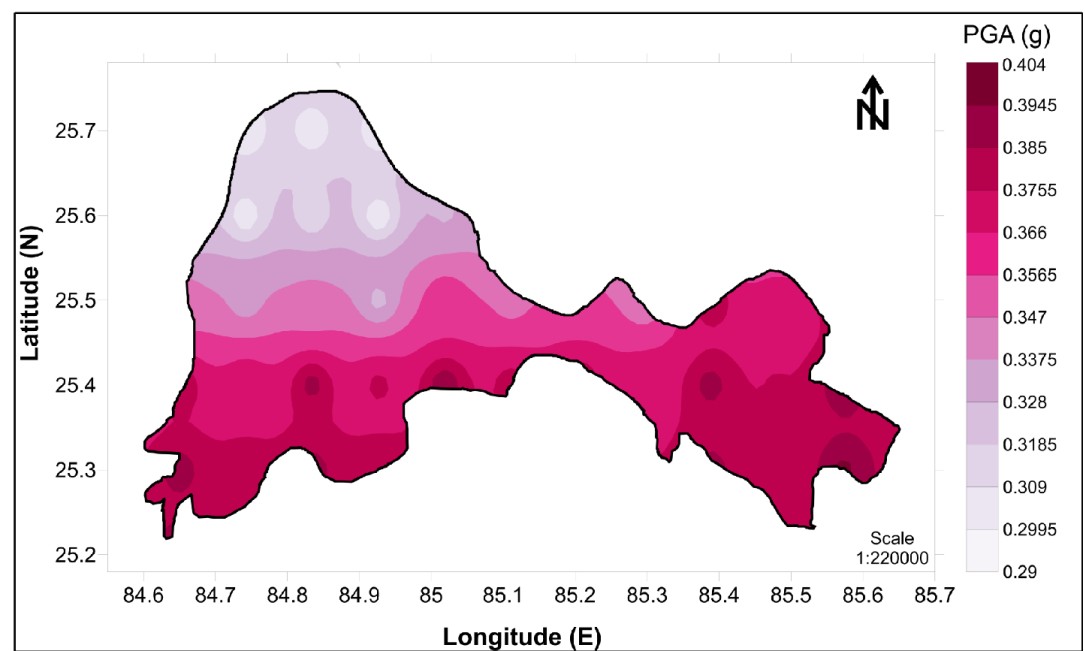

Figure 11 (a): PSHA map for Patna urban centre for 2 % probability of exceedence in 50 years
using areal seismic zone



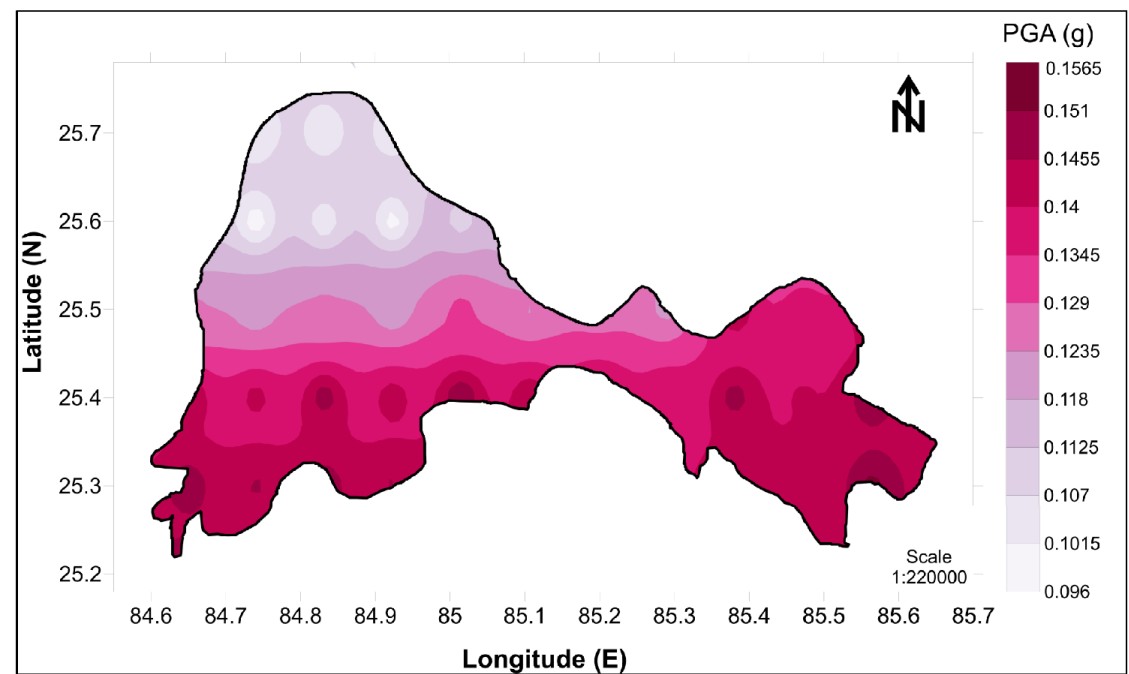

Figure 11 (b): PSHA map for Patna urban centre for 10 % probability of exceedence in 50 years using areal seismic zone





Figure 12: Hazard curve at Patna district centre for different periods using Frankel approach (1995)





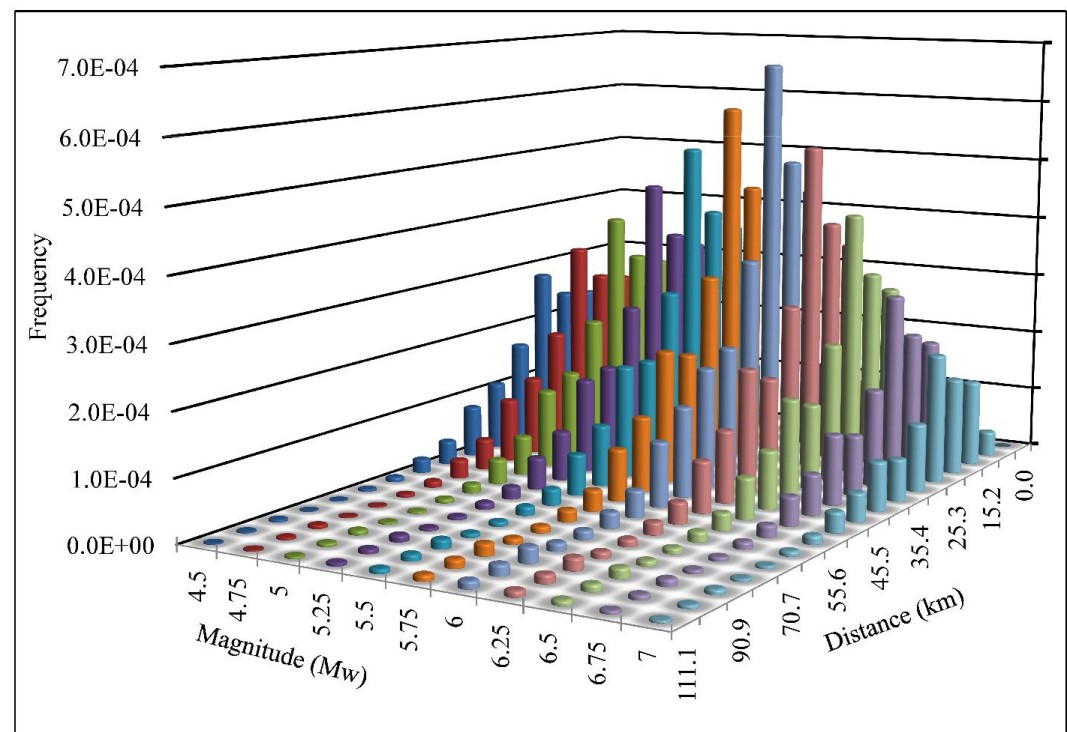

Figure 13 (a): Deaggregation of hazard value at Patna at bed rock at PGA for 2 % probability of exceedence in 50 years using Frankel approach (1995)




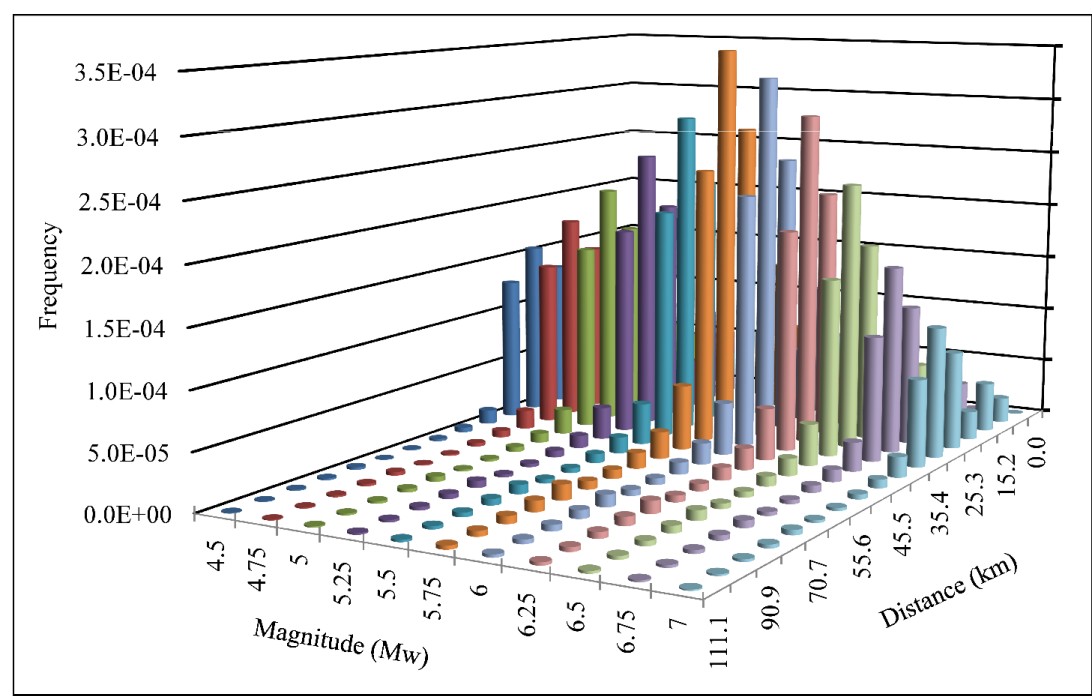

Figure 13 (b): Deaggregation of hazard value at Patna at bed rock at PGA for 10 % probability of exceedence in 50 years using Frankel approach (1995)





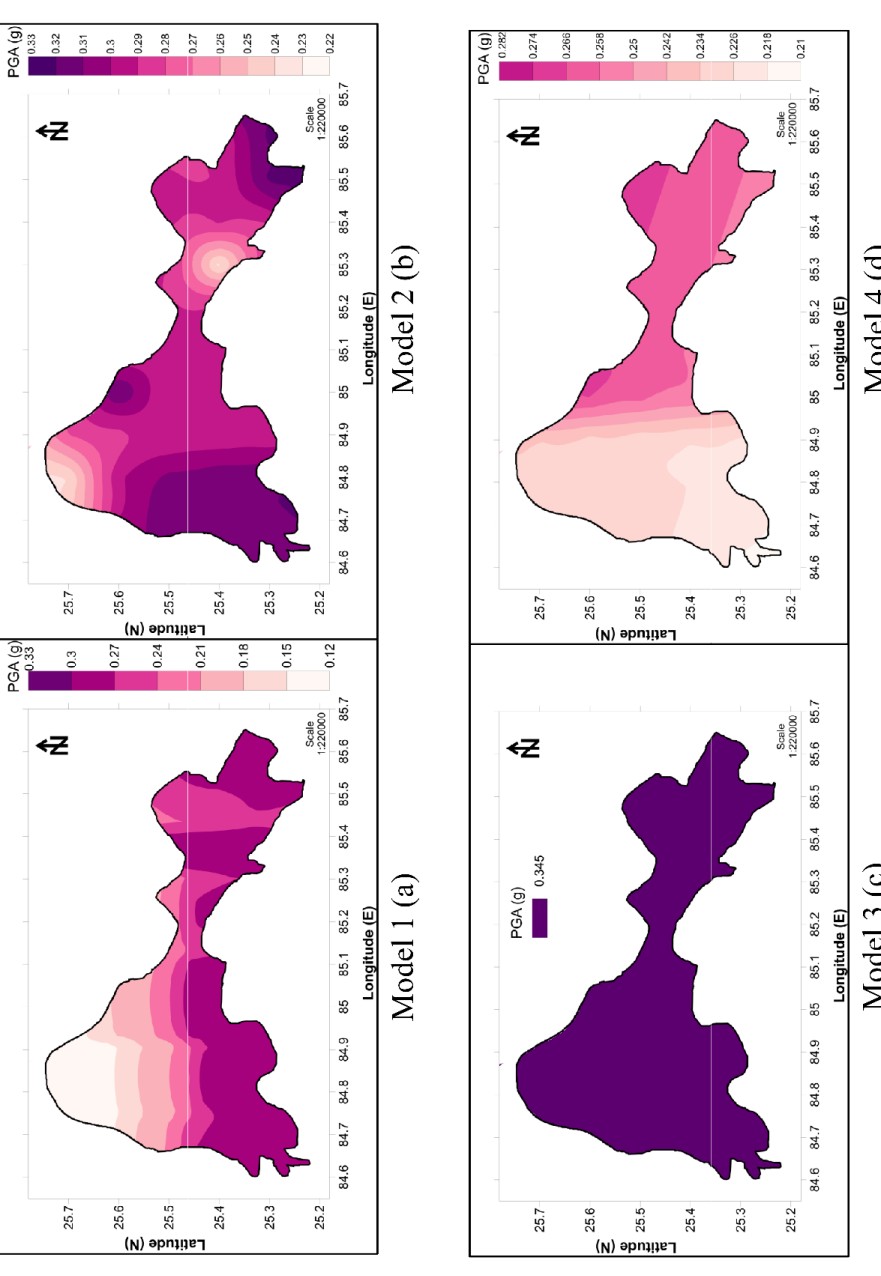

Figure 14: PGA map of Patna SSA considering 2% probability in 50 years for model 1 (a), model 2 (b), model 3 (c) and model 4 (d) using Frankel approach (1995)




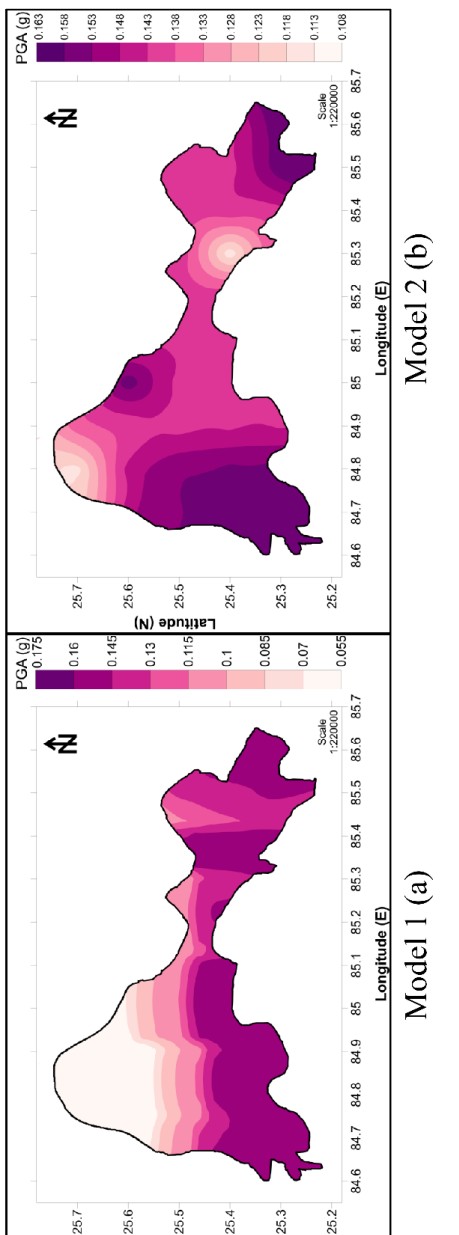

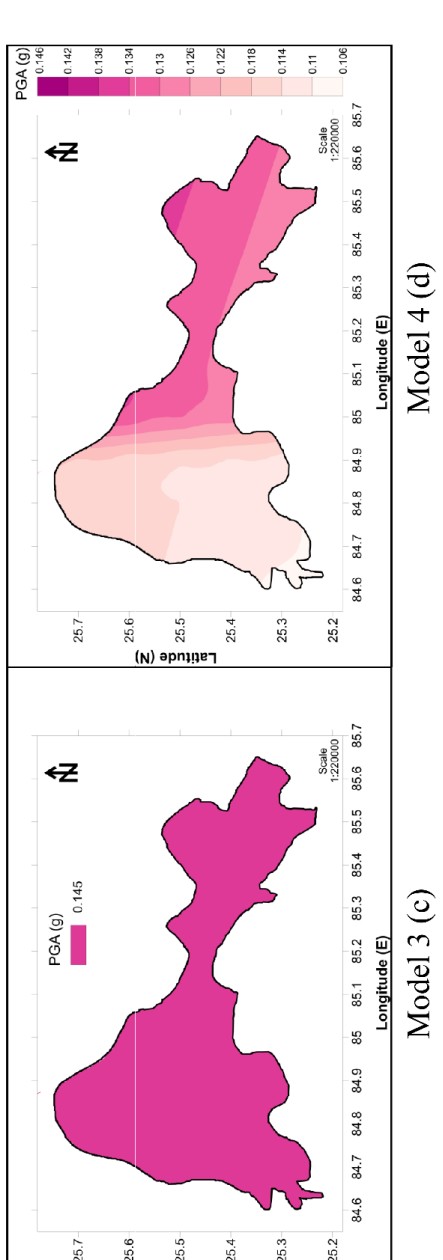

Figure 15: PGA map of Patna SSA considering 10% probability in 50 years for model 1 (a), model 2 (b), model 3 (c) and model 4 (d) using Frankel approach (1995)





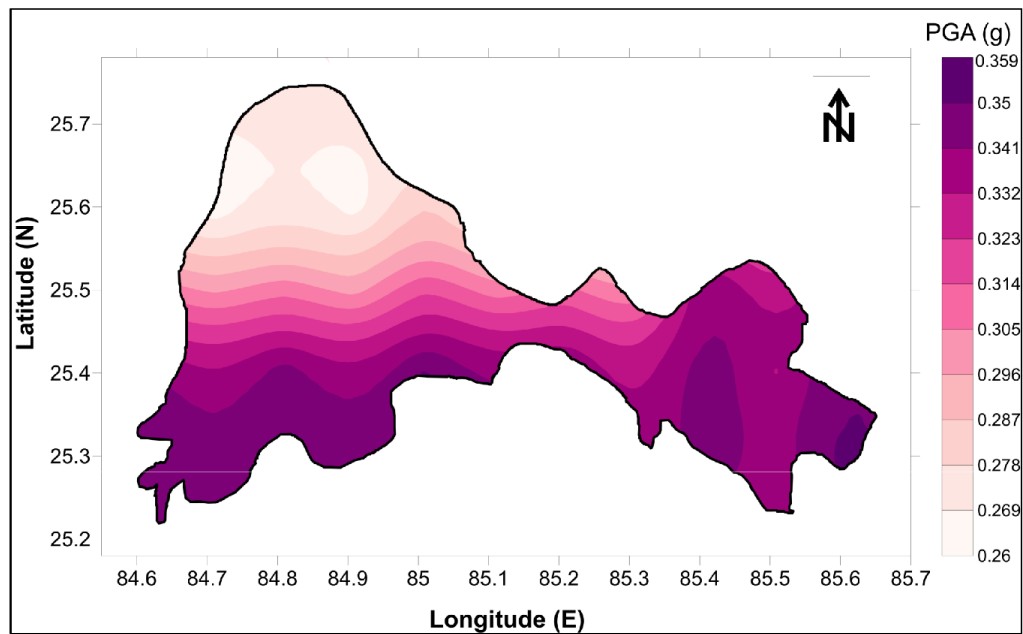

Figure16 (a): Weighted mean PGA map of Patna SSA for 2% probability of exceedence in 50 years using Frankel approach (1995)



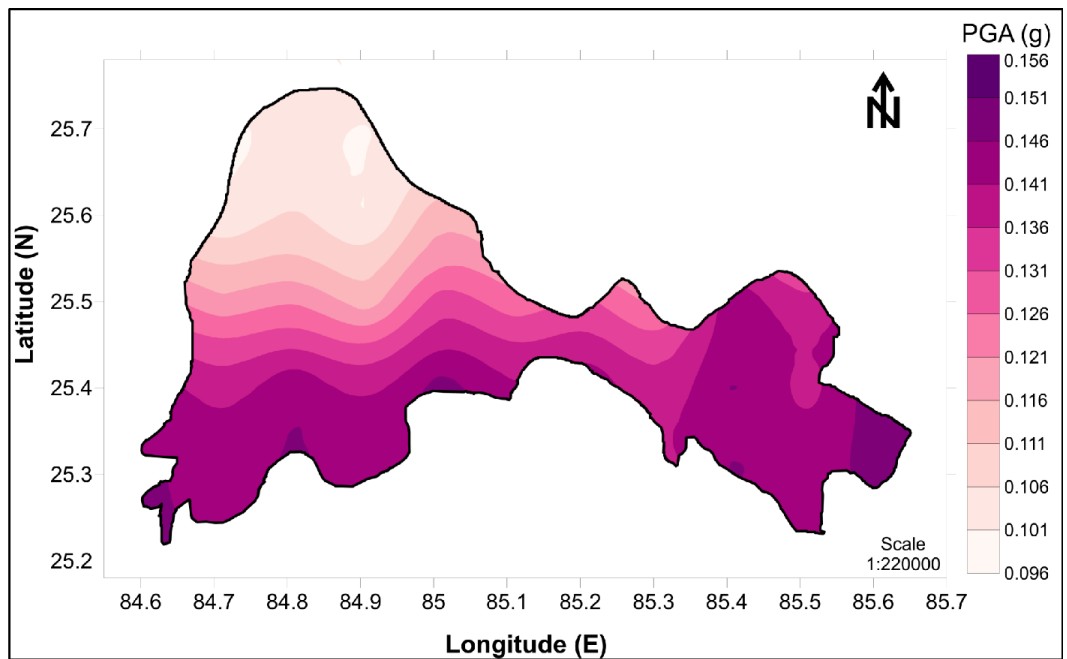

Figure 16 (b): Weighted mean PGA map of Patna SSA for 10% probability of exceedence in 50 years using Frankel approach (1995)



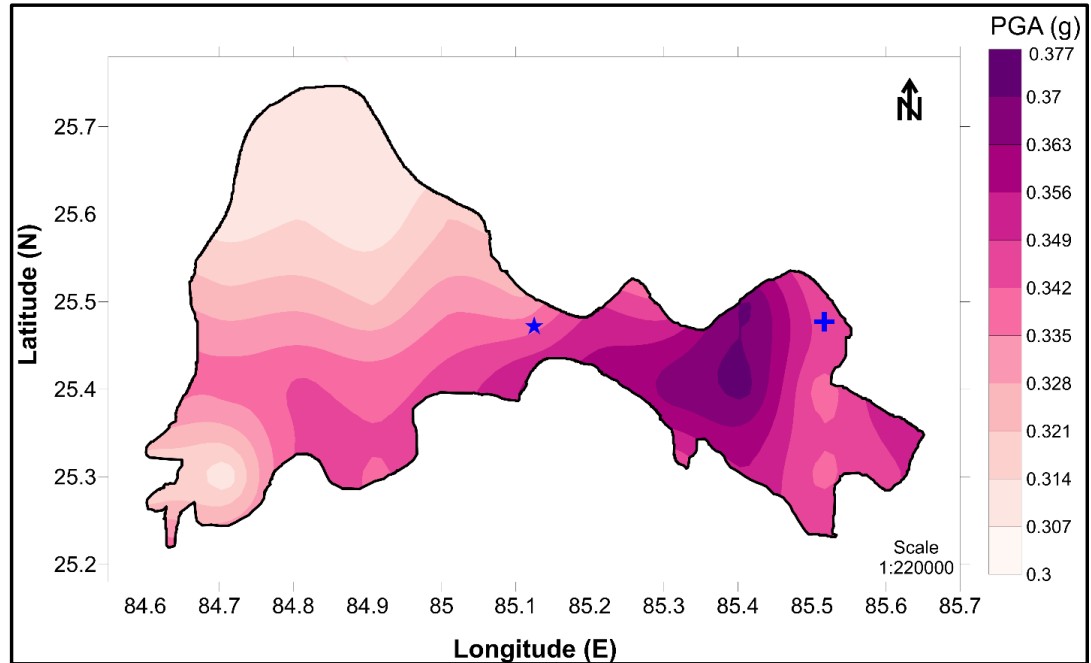

**Figure 17 (a): Final seismic hazard map of Patna SSA for 2% probability of exceedence in 50 years using Logic tree approach**



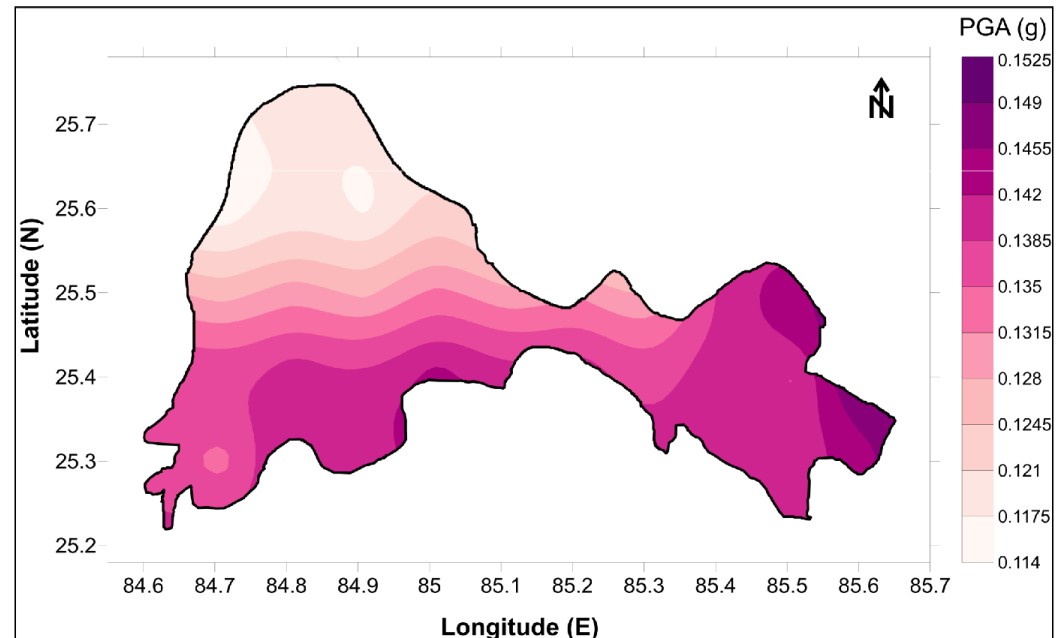

Figure 17 (b): Final seismic hazard map of Patna SSA for 10% probability of exceedence in 50 years using Logic tree approach



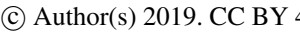


Figure 18: Final seismic hazard map of Patna SSA for (a) 2% and (b) 10% probability of exceedence in 50 years at 0.2 s respectively and (c) 2% and (d) 10% probability of exceedence in 50 years at 1 s respectively using Logic tree approach





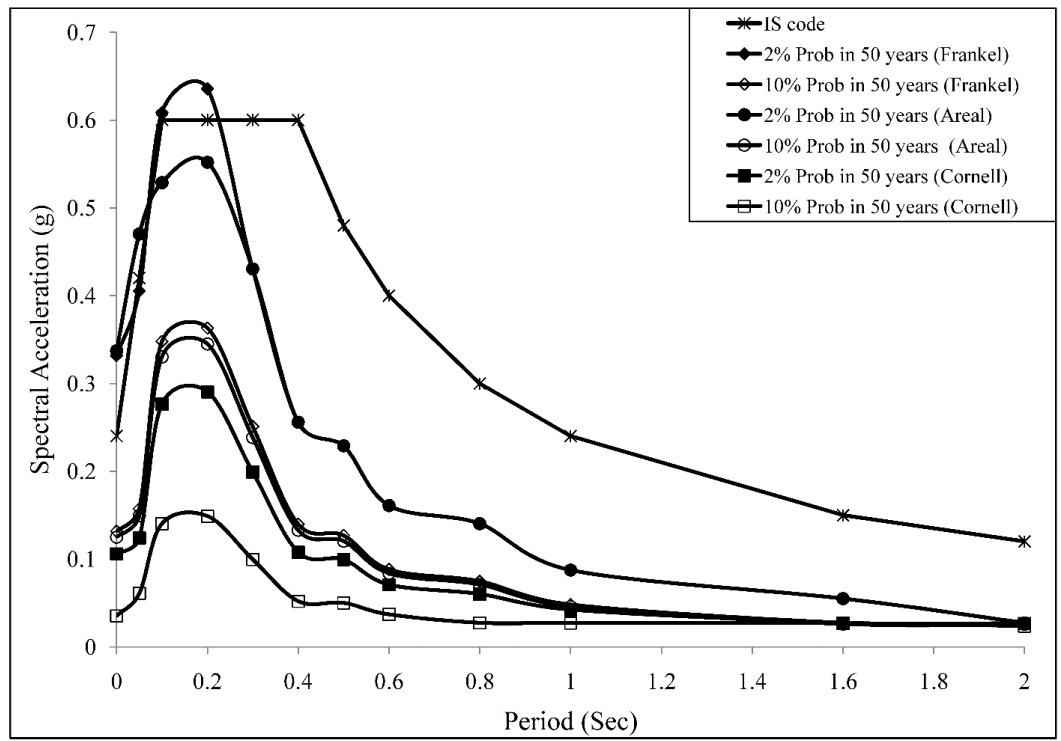

Figure 19 (a): Design spectrum for Patna for 5% damping from 2% and 10% probability of
exceedence in 50 years and IS 1893 (2002) at centre of the city (marked in Figure 17 a)





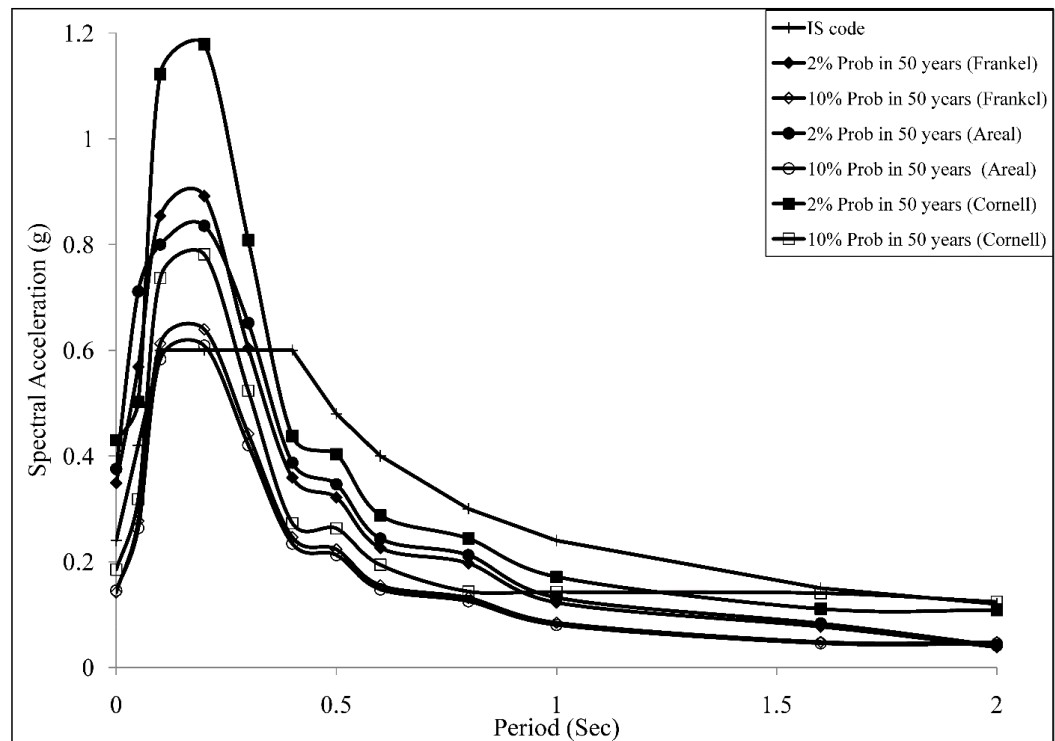

Figure 19 (b): Design spectrum for Patna for 5% damping from 2% and 10% probability of
exceedence in 50 years and IS 1893 (2002) at north eastern part of the city (marked in Figure 17
a)