# Peer review of "Probabilistic seismic hazard analysis using logic tree approach-Patna District (India)"

_Natural Hazards and Earth System Sciences, 2018_

## Referee Comment (RC1) · Gullu (Referee) · 3 Feb 2019

Some comments and criticisms for an application of logic tree approach (Probabilistic seismic hazard analysis using logic tree approach-Patna District, India)

Hamza Güllü*

Department of Civil Engineering, University of Gaziantep, 27310, Gaziantep, Turkey

* Corresponding Author Tel: +90.342. 317 24 33 Email: hgullu@gantep.edu.tr

Abstract

In the article of "Probabilistic seismic hazard analysis using logic tree approach-Patna District (India)" (Nat. Hazards Earth Syst. Sci. Discuss., https://doi.org/10.5194/nhess-

2018-328) studied by Anbazhagan et al., a popular tool called the logic tree approach is employed for seismic hazard analysis of Patna District, India. Despite being an extensive study, it is observed that the logic tree application needs to be more informative about the weighting factors of terminal branches and selection of attenuation equations. This discussion mainly aims to present some comments and criticisms for some clarifications of the logic tree application.

Key words: Logic tree, weighting factors, seismic hazard analysis, attenuation equation.

Due to its capability of combination of multiple models alternatively, the logic tree approach employed in the article is of scientifically significance that practically offers a solution for the issues of the seismicity of the region (Patna District, India). However, the following technical points are the comments that could be queried for the application of logic tree approach in the study.

1) In the logic tree approach, the seismic hazard analysis is carried out by the combination of models and/or parameters constructed with each terminal branch regarding with weighting factors. However, for construction of logic tree branches with the weightings of models, it appears that the criteria are lack and/or not clear in the article. They are the questions that what are the experimenter's (authors') concerns (issues) in practice and what are the expert's recommendations about the seismicity of the region. As a consequence, without accounting the weighting factors realistically, it is not possible to obtain a realistic result of seismic hazard analysis using the logic tree (Gullu and Iyisan, 2016).

2) One of the power utilities of the logic tree comes from its relatively less effort compared to the conventional seismic hazard methodologies. It is important to note that using the logic tree with the judged weighting factor requires a calculation effort that dramatically increases with increased branches (Bommer et al., 2005; Sabetta et al., 2005). Thus, in order for preventing the troubles from the increased branches dur-

ing estimations, the branches with slight differences are strongly recommended to be avoided (Bommer et al., 2005). Hence, readers of the article should be informed whether the authors avoided from similar nodes in the logic tree branches. Again, this specifically requires presentation of selection criteria of weighting factors in detail.

3) Past works (Sabetta et al. 2005; Scherbaum and Kühn, 2011) indicate that selection of attenuation models (i.e., ground motion prediction equations) is much important for seismic hazard analysis using the logic tree approach. Moreover, their selection for the seismic hazard assessment has a greater impact than expert's judgments for the weightings of the logic tree branches. In order to provide a consistency within a probabilistic framework, it is proposed (Scherbaum and Kühn, 2011) that the weight factors in attenuation equations are assigned in a sequential manner (such that if the first equation of three selected gains a weight of 0.6, then the remaining equations as sum must be 0.4). Consequently, the study in the article requires being more informative about how the authors assigned the weights of their selected attenuation equations into account of logic tree frame.

4) In the article, the authors perform seismic hazard estimations by Frankel approach as well as the logic tree. The logic tree estimations should principally show the whole terminal branches (i.e., combinations of all possible models), not sub-branches. However, the study is not convincing that how the authors can compare the logic tree's responses with the ones of its sub-branch of Frankel approach. This makes confusing about the estimation by Frankel approach whether it is estimated using sub-branches of logic tress or using its relevant formula.

The clarifications of the concerns above would contribute to better understanding the ability of logic tree alternative to other methodologies.

References

Bommer, J.J., Scherbaum, F., Bungum, H., Cotton, F., Sabetta, F., and Abrahamson, N.A.: On the use of logic trees for ground-motion prediction equations in seismic haz-

ard analysis. Bulletin of the Seismological Society of America 95(2), 377–389, 2005. Gullu, H. and Iyisan, R.: A Seismic hazard study through the comparison of ground motion prediction equations using the weighting factor of logic tree. Journal of Earthquake Engineering 20(6), 861-884, 2016. Sabetta, F., Lucantoni, A., Bungum, H. and Bommer, J.J.: Sensitivity of PSHA results to ground-motion prediction relations and logic-tree weights. Soil Dynamics and Earthquake Engineering 25, 317-329, 2005. Scherbaum, F. and Kühn, N.: Logic tree branch weights and probabilities: summing up to one is not enough. Earthquake Spectra 27, 1237–1251, 2011.

---

## Referee Comment (RC2) · Anonymous Referee #2 · 6 Feb 2019

The Manuscript entitled "Probabilistic seismic hazard analysis using logic tree approach- Patna District (India)" presents a comprehensive PSHA study for one specific region in north India. Authors employ different alternatives for main PSHA-analyse components including, e.g., $M_c$, maximum magnitude, GMPE-set, zonation model, etc. to populate the epistemic logic tree. The study is confident, uses extensive local sources dataset and employs up-to-date PSHA analytical tools incorporated into the logic tree approach to treat the epistemic uncertainty.

In general, I would recommend publishing present study in NHESS. Nevertheless, I would recommend "major revision" because of the two issues. Both issues deal with the art of presentation, so, I think, Authors could easily accommodate them. First- the manuscript has too many figures in the results section, namely 23! Some of them could

be combined together into one plot. For example figures presenting PGA maps for the three approaches: 'classical', 'areal seismic zone' and 'Frankel' (Fig. 8a, 11a, 16a). Same for the deaggregation diagrams, and so on. Such a combination, if possible, would make presentation more structured and comparison between methods more evident. Alternatively, Authors may think of moving some figures into the supplementary material.

The second issue is writing style. English is generally OK, but the writing style is somewhat sloppy. Especially in the beginning of the manuscript. Please read thoroughly statement-by-statement and put attention at clarity and correctness of the text. To avoid dubious statements like that on Page 2, Lines 10-11.

Following are some more specific comments (referred by page and line number).

1-17: tsunami

1-18: Triggering tsunamis is nothing to do with ground shaking because tsunamis respond to residiual, static deformation of the seabed, not to PGV or PGA.

1-20: "subduction"

1-20: I am not sure if you can call the India-Eurasia collision as "subduction zone" because the latter term commonly implies subduction of the oceanic lithosphere whereas in this case we actually have continent-to-continent collision.

2-6: Does aleatoric uncertainty include "randomness of ground motion prediction"? GMPE's are derived by people, not by nature. Maybe, better to say that it includes randomness of wave propagation and site amplification?

2-11: I do not see the logical connection between the sentence starting with "Generally, ground motion. . .." and the next one. Logic tree is used to quantify all kinds of epistemic uncertainty, not only that related to GMPE's. Please consider re-formulating these paragraph.

2-15: if weight is assigned, we cannot speak about "qualitative" assessment any more

2-21: "As per Bilham" – what is "per"?

2-28: "determined weighted mean"?

2-31: "viz." ?

3-7: what is "SSA". Define explicitly before using abbreviation for the first time.

3-8: an area cannot have only one single value of lon and lat. A point can, area – not.

3-10: give reference to Figure 1 in the beginning of Patna region description

Figure 1: source labels not readable I suggest to add a supplementary table describing individual faults. Or, alternatively, to extend Table S1 with additional parameters like position, rupture length.

3-16/17: redundancy

3-28: this sentence looks redundant. The whole paragraph is better to move to the beginning of the current chapter.

4-21: it is still worth to provide GR-expression with 'a' and 'b' parameters

Seismicity parameters 'a' and 'b' are discussed in both Sections 3.1 and 3.2. That is why present Section titles look somewhat misleading. Consider renaming these sections, for example, according to the derivation approach: period of completeness (3.1) vs magnitude of completeness (3.2).

5-13: why M4.5 was finally accepted as Mc? This statement comes into contradiction with following statements where Authors accept M6-model to be their reference model. M6 has different Mc values for the two regions.

General Remark to Section 3.2: Authors employ 9 different methods to estimate 'a', 'b', and Mc. But finally accept only one model, M6, giving the corresponding logic tree node weight = 0.5. That means all other models were given zero weights despite some

of them (M1,3,5) show results similar to M6. Authors should more clear justify why they do neglect all other 8 models.

9-29: vulnerable?

---

## Referee Comment (RC3) · Anonymous Referee #3 · 7 Feb 2019

Journal: NHESS Title: Probabilistic seismic hazard analysis using logic tree approach – Patna District (India) Author(s): Panjamani Anbazhagan et al. MS No.: nhess-2018-328 The article titled "Probabilistic seismic hazard analysis using logic tree approach-Patna District (India)" utilize logic tree technique to conduct PSHA study for Patna District, India. Authors employ different branches in the logic tree for PSHA calculations to handle the epistemic uncertainties. Although the work is extensive and the exerted efforts are great, this paper still needs many clarifications so it can be accepted for publication. It is not well organized and in many parts, it is non-properly sequenced with non-threaded paragraphs, leaving the reader confused and suffering to catch the idea. The English language of the paper is poor and negatively affects the understanding of many paragraphs. English needs to be revised critically. Abbreviations should be

mentioned at its first appearance. Avoid to use the same abbreviation for two different terms (e.g. SA is used for spectral acceleration and for study area). What are SSA, MBT, MCT, S60, ....etc. All abbreviations should be defined at their first appearance in the text. All localities, faults and geological structures mentioned in the manuscript should be shown on maps. I could not appropriately follow the seismotectonic part of the area due to the lacking of such illustrations. Now let me provides some other comments in a sequential form: Introduction Page 1, lines 20-21: Which gap? Please provide more explanation. Page 2, lines 3-5: Very accurate sentence, but nothing is carried out in the end. Why this sentence is written here? Page 2, line 27: I could not understand "Maximum magnitude has been determined weighted mean using increment . . . . . .." Geology, Seismotectonics and seismicity of the study area (SA) Page 3, line 8: coordinates here are for a point, it is not for an area. Page 3, line 29: "and published literatures" give references. Page 4, lines 1-3: Authors should show the priority scheme in selecting the earthquake from each data base. I mean if the same earthquake is available in more than one database, which one will selected? Which magnitude scale from which database has the first priority and which has the second and so on? Is the same magnitude scale for the same earthquake at different database yield the same value? All the above queries should be clarified in detail. Please show the start and end time of the catalogue to be able to assess its reliability. Page 4, lines 15-18: Please revise the earthquake numbers in each magnitude range as their sum should be 818 as mentioned in Page 4 line 9. a and b parameters This is the most confusing part of the manuscript. In this section the a and b values are calculated for two regions (I and II). What is the role of these two area and their seismicity parameters in the hazard calculations. The classical method used 178 seismic sources and the zoneless method used 7 area seismic zones. Why this is interfered in the current study. Secondly, the magnitude of completeness should be calculated before evaluating the seismicity parameters as GR parameters should use complete data only. Magnitude of completeness Page 5, line 12: This great difference in the Mc values casts doubt on the calculated values. Please explain why different methods have such different

outputs. Also justify the great difference in a and b values in lines 17-19. B values of 0.149 and 0.176 are not physically accepted. Again, it is not clear how the authors used the a and b values shown in this section in the hazard calculations? Page 5, line 32: "based on b values" to add 0.5 based on b value, b value should range between -0.9 and -1.0, which is not the case here. Maximum magnitude estimation (Mmax) The authors used the region specific rupture technique to calculate M max and provide it the maximum weight. The technique depends on the ratio between the rupture length and the total fault length. My questions are: 1- Is the seismic record enough to be sure about the above ratio? The answer is definitely NO as the authors themselves clarified when they justify the use of zoneless method, stating that "many sources given in Figure 1 are not well studied to prove its seismic activity". This raises great uncertainty on the maximum magnitude calculated for these seismic sources. 2- Is there any possibility to rupture the entire fault length in one earthquake? Recent studies suppose that the entire fault length will be ruptured in one earthquake when calculating the maximum earthquake. 8.1 classical approach Page 9, line 27: Authors used 178 seismic sources. The seismicity of many of these faults are not well studied. It is not clear how the seismicity parameters are calculated for each single source. It is well known that GR model cannot be used to calculate a and b values for single faults. Slip rate could be used but with many not well studied sources, the results should be at least uncertain. Using logic tree does not mean ignoring use the right input parameters for each method. Zoneless approach Page 10, line27: use return period instead of "frequency of exceedance" Four models (figure 4) using zoneless approach (Frankel, 1995) Page 11, line 15: the return period 85 years (of what? This is most probably PGA) Page 11, line 19: From which model the deaggregation plot is calculated? Or the authors used weighted deaggregation values based upon the weighs given for each of the four models. This should be very clear. Authors should explain why the results of the two methods are completely different in terms of hazard values and terms of the change in the spatial distribution (many low hazard areas in one method show very high hazard in the other method). This should be justified, as it is not enough to say for this the

logic tree is created. A mistake could be done in the calculation or a method is not adequate for the region. Therefore, it is better to justify the use of zoneless methods. Page 12, line 5: Please add for 10% probability before "The PGA values" Final hazard map using logic tree Page 12, lines 26-27: As the high hazard values are related to the East and West Patna Fault, then, why the classical hazard values which are more related to the faults show very much less values?? Authors compared their final results with previous studies. I recommend to compare the results of each method with the recent observations and with the previous studies to show a reason why the results are very inconsistent. If the current results are accurate, authors should recommend to change IS 1893 (2002) in Patna as the current hazard values highly exceed its summit. Figure 1 is very unclear and need to be provided in a higher resolution way.

The manuscript should be thoroughly and meticulously revised and minimize self-citations and refer to more original, only essential published articles.

---

## Author Comment (AC1) · 12 Apr 2019

General Comment: - The Manuscript entitled "Probabilistic seismic hazard analysis using logic tree approach- Patna District (India)" presents a comprehensive PSHA study for one specïñĄc region in north India. Authors employ different alternatives for main PSHA-analyses components including, e.g., Mc, maximum magnitude, GMPE-set, zonation model, etc. to populate the epistemic logic tree. The study is conïñĄdent, uses extensive local sources dataset and employs up-to-date PSHA analytical tools incorporated into the logic tree approach to treat the epistemic uncertainty. In general, I would recommend publishing present study in NHESS. Nevertheless, I would recommend "major revision" because of the two issues. Both issues deal with the art of presentation, so, I think, Authors could easily accommodate them. First- the

manuscript has too many figures in the results section, namely23! Some of them could be combined into one plot. For example, figures presenting PGA maps for the three approaches: 'classical', 'areal seismic zone' and 'Frankel' (Fig. 8a, 11a, 16a). Same for the deaggregation diagrams, and so on. Such a combination, if possible, would make presentation more structured and comparison between methods more evident. Alternatively, Authors may think of moving some figures into the supplementary material. The second issue is writing style. English is generally OK, but the writing style is somewhat sloppy. Especially in the beginning of the manuscript. Please read thoroughly statement-by-statement and put attention at clarity and correctness of the text. To avoid dubious statements like that on Page 2, Lines 10-11. Response: - The authors would like to thank the reviewer for his valuable comments which helped us in reviewing the manuscript. As per the suggestion figures have been combined and few has been used as supplementary material. The writing style has been also improved and the manuscript has been checked thoroughly statement-by-statement. Page 2 and line 10-11 has been revised. Change in the manuscript: In the absence of appropriate region-specific models of wave propagation, ground motion prediction models are generally used to determine the hazard value. Comment 1: - 1-17: tsunami Response: - It has been changed in the revised manuscript. Comment 2: - 1-18: Triggering tsunamis is nothing to do with ground shaking because tsunamis respond to residual, static deformation of the seabed, not to PGV or PGA. Response: - Tsunami has been removed in the revised manuscript. Comment 3: - 1-20: "subduction" Response: - It has been changed in the revised manuscript. Comment 4: - 1-20: I am not sure if you can call the India-Eurasia collision as "subduction zone" because the latter term commonly implies subduction of the oceanic lithosphere whereas in this case, we actually have continent-to-continent collision. Response: - The word "subduction zone" has been replaced by "continent-to-continent collision". Change in the manuscript: Besides, many great events (2015, Nepal earthquake) have originated from continental-to-continental collision. Comment 5: - 2-6: Does aleatoric uncertainty include "randomness of ground motion prediction"? GMPE's are derived by people,

not by nature. Maybe, better to say that it includes randomness of wave propagation and site amplification? Response: - It has been changed as per the suggestion. The statement has been changed as follow Change in the manuscript: One is due to randomness of the nature of earthquake, wave propagation, and site amplification named as aleatory uncertainty while other is due to incomplete knowledge of earthquake process named as epistemic uncertainty. Comment 6: - 2-11: I do not see the logical connection between the sentence starting with "Generally, ground motion: : :." and the next one. Logic tree is used to quantify all kinds of epistemic uncertainty, not only that related to GMPE's. Please consider re-formulating these paragraphs. Response: - As per the suggestion this paragraph has been revised. It has been revised as follow Change in the manuscript: Epistemic uncertainty is due to improper knowledge about the process involve in earthquake events and algorithms used to model them. Hence, in this study, logic tree framework has been used to reduce the epistemic uncertainty in the final hazard value calculation. In the absence of appropriate region-specific models of wave propagation, ground motion prediction models are generally used to determine the hazard value. The uncertainty in GMPEs can be reduced by incorporating logic tree in the hazard analysis study. Comment 7: - 2-15: if weight is assigned, we cannot speak about "qualitative" assessment any more Response: - This word has been removed in the revised manuscript. Comment 8: - 2-21: "As per Bilham" – what is "per"? Response: - "As per" has been replaced with "similar to" Comment 9: - 2-28: "determined weighted mean"? Response: - Apology for the typo. This statement has been revised as below Change in the manuscript: Maximum magnitude has been determined using weighted mean considering three methods as increment factor on maximum observed magnitude, Kijko and Sellevoll (1989) and regional rupture characteristics (Anbazhagan et al. 2015b). Comment 10: - 2-31: "viz." ? Response: - "viz." has been replaced by "namely" Comment 11: - 3-7: what is "SSA". Define explicitly before using abbreviation for the first time. Response: - "SSA" is seismic study area and it has been mentioned in the revised manuscript. Comment 12: - 3-8: an area cannot have only one single value of lon and lat. A point

can, area – not. Response: - The statement has been changed as follow Change in the manuscript: The present study area has covered the longitude 84.6-85.65°E and latitude 25.2-25.8°N Comment 13: - 3-10: give reference to Figure 1 in the beginning of Patna region description Figure 1: source labels not readable I suggest adding a supplementary table describing individual faults. Or, alternatively, to extend Table S1 with additional parameters like position, rupture length. Response: - As per the suggestion, the refence of Figure 1 has been given in the beginning and Table S1 has been extended by providing the position (latitude and longitude of the end points), total fault length and rupture length. Comment 14: - 3-16/17: redundancy Response: - As per the suggestion the sentences are moved blow at relevant position. Comment 15: - 3-28: this sentence looks redundant. The whole paragraph is better to move to the beginning of the current chapter. Response: - As per the suggestion the whole paragraph is moved in the beginning of the paragraph. Change in the manuscript: Based on damage distribution map i.e. isoseismal map (1833 Nepal earthquake and 1934 Bihar-Nepal earthquake) and location of Main Boundary Trust, Main Central Trust and Himalayan Frontal Thrust (HFT), a radius of 500 km has been selected for present SSA. The detail study about selecting SA of 500 km is given in Anbazhagan et al. (2015a). Geographical information of India demonstrates that approximately 60 % of the land is highly susceptible to earthquakes (NDMA, 2010). The tectonic feature of SA has been compiled from the Seismotectonic Atlas (SEISAT, 2010) published by the Geological Survey of India (GSI, 2000). The seismotectonic map was developed by considering 500 km radius from Patna district boundary by considering linear sources (faults and lineaments) from SEISAT and published literatures (e.g. NDMA, 2010; Nath and Thingbaijam, 2012; Kumar et al., 2013). Separation of MBT and MCT has been done and all the faults along with MBT and MCT have also been numbered. Seismotectonic map for Patna District is shown in Figure 1. A brief description of seismicity and seismotectonics of SSA is given below. Comment 16: - 4-21: it is still worth to provide GR-expression with 'a' and 'b' parameters Seismicity parameters 'a' and 'b' are discussed in both Sections 3.1 and 3.2. That is why present Section

titles look somewhat misleading. Consider renaming these sections, for example, according to the derivation approach: period of completeness (3.1) vs magnitude of completeness (3.2). Response: - Both the sections have been renamed as per the suggestion Comment 17: - 5-13: why M4.5 was finally accepted as Mc? This statement comes into contradiction with following statements where Authors accept M6-model to be their reference model. M6 has different Mc values for the two regions. Response: - Apology for the same. This statement has been removed as it's a typo error. Comment 18: - General Remark to Section 3.2: Authors employ 9 different methods to estimate 'a', 'b', and Mc. But finally accept only one model, M6, giving the corresponding logic tree node weight = 0.5. That means all other models were given zero weights despite some of them (M1,3,5) show results similar to M6. Authors should clearer justify why they do neglect all other 8 models. Response: - Nine methods have been used to check the variability in 'a', 'b', and Mc for the same study area. However as per Boomer et al. (2005) calculation effort increases dramatically with the inclusion of more branches in the logic tree. Therefore, Bommer et al. (2005) suggested avoiding using branches with slight differences between the options, in cases when those options result in very similar nodes. Hence only M6 has been used as M6 method is capable for M_c calculation as it synthetically maximises the available data and stabilises the M_c value. Change in the manuscript: According to Boomer et al. (2005) calculation effort increases dramatically with the inclusion of more branches in the logic tree. Therefore, Bommer et al. (2005) suggested avoiding using branches with slight differences between the options, in cases when those options result in very similar nodes. Hence only M6 has been used as M6 method is capable for M_c calculation as it synthetically maximises the available data and stabilises the M_c value. Comment 19: - 9-29: vulnerable? Response: - Apology for the typo. This word has been replaced. ====== END ======

Please also note the supplement to this comment:
https://www.nat-hazards-earth-syst-sci-discuss.net/nhess-2018-328/nhess-2018-328-

[Figure]

AC1-supplement.pdf

---

## Author Comment (AC2) · 12 Apr 2019

General Comment: - Abstract In the article of "Probabilistic seismic hazard analysis using logic tree approach-Patna District (India)" (Nat. Hazards Earth Syst. Sci. Discuss., https://doi.org/10.5194/nhess 2018-328) studied by Anbazhagan et al., a popular tool called the logic tree approach is employed for seismic hazard analysis of Patna District, India. Despite being an extensive study, it is observed that the logic tree application needs to be more informative about the weighting factors of terminal branches and selection of attenuation equations. This discussion mainly aims to present some comments and criticisms for some clarifications of the logic tree application. Key words: Logic tree, weighting factors, seismic hazard analysis, attenuation equation. Due to its capability of combination of multiple models alternatively, the logic

tree approach employed in the article is of scientifically significance that practically offers a solution for the issues of the seismicity of the region (Patna District, India). However, the following technical points are the comments that could be queried for the application of logic tree approach in the study. Response: - The authors would like to thank the reviewer for his/her valuable time for reviewing the manuscript. The following are the detailed response to the comments. Comment 1: - In the logic tree approach, the seismic hazard analysis is carried out by the combination of models and/or parameters constructed with each terminal branch regarding with weighting factors. However, for construction of logic tree branches with the weightings of models, it appears that the criteria are lack and/or not clear in the article. They are the questions that what are the experimenter's (authors') concerns (issues) in practice and what are the expert's recommendations about the seismicity of the region. As a consequence, without accounting the weighting factors realistically, it is not possible to obtain a realistic result of seismic hazard analysis using the logic tree (Gullu and Iyisan, 2016). Response: - The questions that what are the experimenter's concerns in practice and what are the expert's recommendations about the seismicity of the region is also explained in the revised article. In the revised manuscript, the construction of logic tree and the weighting of the different branches of the logic tree has been explained at different places. Change in the manuscript: Patna district lies near to the seismically active Himalayan belt and on the deep deposits of the Indo-Gangetic basin (IGB). It is also surrounded by various active ridges as Monghyr-Saharsa Ridge Fault many active tectonic features such as Munger-Saharsa-Ridge Fault, and active faults such as East Patna Fault or West Patna Fault. These faults are acknowledged as transverse faults, and the occurrence of seismic events is due to stimulus of fluvial dynamics in the North Patna plains transverse faults (Valdiya1976; Dasguptaet al.1987). According to Banghar (1991) the East Patna Fault is one of the active faults in the study area and its interaction with Himalayan Frontal Thrust is characterized by a cluster of earthquakes. Dasgupta et al. (1993) accounted that all other faults between Motihari and Kishanganj city have the same possibility of seismic hazard as they form a part of related fault

system. Comment 2: - One of the power utilities of the logic tree comes from its relatively less effort compared to the conventional seismic hazard methodologies. It is important to note that using the logic tree with the judged weighting factor requires a calculation effort that dramatically increases with increased branches (Bommer et al., 2005; Sabetta et al., 2005). Thus, in order for preventing the troubles from the increased branches during estimations, the branches with slight differences are strongly recommended to be avoided (Bommer et al., 2005). Hence, readers of the article should be informed whether the authors avoided from similar nodes in the logic tree branches. Again, this specifically requires presentation of selection criteria of weighting factors in detail. Response: - In the present study, the weight factor for different GMPEs has been calculated using the log likelihood values, which is explained in the manuscript. No such branch having with slight differences in weights have been observed in the present study. Change in the manuscript: It is necessary here to note that the experimenters performing for the seismic hazard assessment using weighting factor may lead to complication in the calculations with the inclusion of different branches. To prevent this trouble, Bommer et al. (2005) suggested avoiding using the branches having slightly differences between the options that it carries, in cases when those options result in very similar nodes. Therefore, when selecting the weighting factors in the logic tree in this study, the cases contrasting (or different) with each other as much as possible have been taken into consideration. Comment 3: - Past works (Sabetta et al. 2005; Scherbaum and Kühn, 2011) indicate that selection of attenuation models (i.e., ground motion prediction equations) is much important for seismic hazard analysis using the logic tree approach. Moreover, their selection for the seismic hazard assessment has a greater impact than expert's judgments for the weightings of the logic tree branches. In order to provide a consistency within a probabilistic framework, it is proposed (Scherbaum and Kühn, 2011) that the weight factors in attenuation equations are assigned in a sequential manner (such that if the first equation of three selected gains a weight of 0.6, then the remaining equations as sum must be 0.4). Consequently, the study in the article requires being more

informative about how the authors assigned the weights of their selected attenuation equations into account of logic tree frame. Response: - We agreed with the reviewer, in the present study the weights have been assigned in the sequential manner. This has been already explained in the revised manuscript with proper references. Change in the manuscript: Scherbaum and Kühn (2011) showed the importance of weight treatments through the logic tree approach as probabilities instead of simply as generic quality measures of attenuation equations, which are subsequently normalized. They also indicated the risk of independently assigning of grades by different quality criteria, which could result in an apparent insensitivity to the weights. In order to provide the consistency with a probabilistic framework, they proposed assigning the weight factors in a sequential manner, which is used in the present study. Comment 4: - In the article, the authors perform seismic hazard estimations by Frankel approach as well as the logic tree. The logic tree estimations should principally show the whole terminal branches (i.e., combinations of all possible models), not sub-branches. However, the study is not convincing that how the authors can compare the logic tree's responses with the ones of its sub-branch of Frankel approach. This makes confusing about the estimation by Frankel approach whether it is estimated using sub-branches of logic tress or using its relevant formula. Response: - In the present study, the hazards values are calculated using the Frankel approach considering the four models proposed by Frankel (1995). Further the final map developed using Frankel (1995) has been weighted and combined with the areal seismic sources to calculate the hazard values using the zoneless approach.

Please also note the supplement to this comment:
https://www.nat-hazards-earth-syst-sci-discuss.net/nhess-2018-328/nhess-2018-328-AC2-supplement.pdf
* * *
[Figure]

**Supplement:**

**Detailed Response Letter for NHESS-2018-328 Review Comments**

**Probabilistic seismic hazard analysis using logic tree approach- Patna District (India)**

Panjamani Anbazhagan[1], Ketan Bajaj[1], Karanpreet Matharu[1], Sayed S. R. Moustafa[2], Nassir S. N. Al-Arifi[2]

**Response to All the Reviewers Comments**

We thank all reviews and editor for their valuable to time to review manuscript and give valuable suggestions to improve the same. Most of the comments are suggestions to improve the current version of the manuscript, which will be incorporated in the revised manuscript. Few clarifications are raised; we have given our response for the same below and also highlighted that respective points will be added in the revised manuscript. As editor informed only prepare response to reviewers comments, we are not submitting revised manuscript now, But we have given revised text that will be added in the revised version of manuscript.

**Response to Reviewer 1**

**General Comment: -** Abstract In the article of "Probabilistic seismic hazard analysis using logic tree approach-Patna District (India)" (Nat. Hazards Earth Syst. Sci. Discuss., https://doi.org/10.5194/nhess 2018-328) studied by Anbazhagan et al., a popular tool called the logic tree approach is employed for seismic hazard analysis of Patna District, India. Despite being an extensive study, it is observed that the logic tree application needs to be more informative about the weighting factors of terminal branches and selection of attenuation equations. This discussion mainly aims to present some comments and criticisms for some clarifications of the logic tree application.

Key words: Logic tree, weighting factors, seismic hazard analysis, attenuation equation.

Due to its capability of combination of multiple models alternatively, the logic tree approach employed in the article is of scientifically significance that practically offers a solution for the issues of the seismicity of the region (Patna District, India). However, the following technical points are the comments that could be queried for the application of logic tree approach in the study.

**Response: -** The authors would like to thank the reviewer for his/her valuable time for reviewing the manuscript. The following are the detailed response to the comments.

**Comment 1: -** In the logic tree approach, the seismic hazard analysis is carried out by the combination of models and/or parameters constructed with each terminal branch regarding with

weighting factors. However, for construction of logic tree branches with the weightings of models, it appears that the criteria are lack and/or not clear in the article. They are the questions that what are the experimenter's (authors') concerns (issues) in practice and what are the expert's recommendations about the seismicity of the region. As a consequence, without accounting the weighting factors realistically, it is not possible to obtain a realistic result of seismic hazard analysis using the logic tree (Gullu and Iyisan, 2016).

**Response: -** The questions that what are the experimenter's concerns in practice and what are the expert's recommendations about the seismicity of the region is also explained in the revised article. In the revised manuscript, the construction of logic tree and the weighting of the different branches of the logic tree has been explained at different places.

**Change in the manuscript:** Patna district lies near to the seismically active Himalayan belt and on the deep deposits of the Indo-Gangetic basin (IGB). It is also surrounded by various active ridges as Monghyr-Saharsa Ridge Fault many active tectonic features such as Munger-Saharsa-Ridge Fault, and active faults such as East Patna Fault or West Patna Fault. These faults are acknowledged as transverse faults, and the occurrence of seismic events is due to stimulus of fluvial dynamics in the North Patna plains transverse faults (Valdiya1976; Dasguptaet al.1987). According to Banghar (1991) the East Patna Fault is one of the active faults in the study area and its interaction with Himalayan Frontal Thrust is characterized by a cluster of earthquakes. Dasgupta et al. (1993) accounted that all other faults between Motihari and Kishanganj city have the same possibility of seismic hazard as they form a part of related fault system.

**Comment 2: -** One of the power utilities of the logic tree comes from its relatively less effort compared to the conventional seismic hazard methodologies. It is important to note that using the logic tree with the judged weighting factor requires a calculation effort that dramatically increases with increased branches (Bommer et al., 2005; Sabetta et al., 2005). Thus, in order for preventing the troubles from the increased branches during estimations, the branches with slight differences are strongly recommended to be avoided (Bommer et al., 2005). Hence, readers of the article should be informed whether the authors avoided from similar nodes in the logic tree branches. Again, this specifically requires presentation of selection criteria of weighting factors in detail.

**Response: -** In the present study, the weight factor for different GMPEs has been calculated using the log likelihood values, which is explained in the manuscript. No such branch having with slight differences in weights have been observed in the present study.

**Change in the manuscript:** It is necessary here to note that the experimenters performing for the seismic hazard assessment using weighting factor may lead to complication in the calculations with the inclusion of different branches. To prevent this trouble, Bommer et al. (2005) suggested avoiding using the branches having slightly differences between the options that it carries, in cases when those options result in very similar nodes. Therefore, when selecting

**Comment 3: -** Past works (Sabetta et al. 2005; Scherbaum and Kühn, 2011) indicate that selection of attenuation models (i.e., ground motion prediction equations) is much important for seismic hazard analysis using the logic tree approach. Moreover, their selection for the seismic hazard assessment has a greater impact than expert's judgments for the weightings of the logic tree branches. In order to provide a consistency within a probabilistic framework, it is proposed (Scherbaum and Kühn, 2011) that the weight factors in attenuation equations are assigned in a sequential manner (such that if the first equation of three selected gains a weight of 0.6, then the remaining equations as sum must be 0.4). Consequently, the study in the article requires being more informative about how the authors assigned the weights of their selected attenuation equations into account of logic tree frame.

**Response: -** We agreed with the reviewer, in the present study the weights have been assigned in the sequential manner. This has been already explained in the revised manuscript with proper references.

**Change in the manuscript:** Scherbaum and Kühn (2011) showed the importance of weight treatments through the logic tree approach as probabilities instead of simply as generic quality measures of attenuation equations, which are subsequently normalized. They also indicated the risk of independently assigning of grades by different quality criteria, which could result in an apparent insensitivity to the weights. In order to provide the consistency with a probabilistic framework, they proposed assigning the weight factors in a sequential manner, which is used in the present study.

**Comment 4: -** In the article, the authors perform seismic hazard estimations by Frankel approach as well as the logic tree. The logic tree estimations should principally show the whole terminal branches (i.e., combinations of all possible models), not sub-branches. However, the study is not convincing that how the authors can compare the logic tree's responses with the ones of its sub-branch of Frankel approach. This makes confusing about the estimation by Frankel approach whether it is estimated using sub-branches of logic tress or using its relevant formula.

**Response: -** In the present study, the hazards values are calculated using the Frankel approach considering the four models proposed by Frankel (1995). Further the final map developed using Frankel (1995) has been weighted and combined with the areal seismic sources to calculate the hazard values using the zoneless approach.

**Response to Reviewer 2**

**General Comment: -** The Manuscript entitled "Probabilistic seismic hazard analysis using logic tree approach- Patna District (India)" presents a comprehensive PSHA study for one specific region in north India. Authors employ different alternatives for main PSHA-analyses components including, e.g., Mc, maximum magnitude, GMPE-set, zonation model, etc. to populate the epistemic logic tree. The study is confident, uses extensive local sources dataset and employs up-to-date PSHA analytical tools incorporated into the logic tree approach to treat the epistemic uncertainty. In general, I would recommend publishing present study in NHESS. Nevertheless, I would recommend "major revision" because of the two issues. Both issues deal with the art of presentation, so, I think, Authors could easily accommodate them. First- the manuscript has too many figures in the results section, namely23! Some of them could be combined into one plot. For example, figures presenting PGA maps for the three approaches: 'classical', 'areal seismic zone' and 'Frankel' (Fig. 8a, 11a, 16a). Same for the deaggregation diagrams, and so on. Such a combination, if possible, would make presentation more structured and comparison between methods more evident. Alternatively, Authors may think of moving some figures into the supplementary material. The second issue is writing style. English is generally OK, but the writing style is somewhat sloppy. Especially in the beginning of the manuscript. Please read thoroughly statement-by-statement and put attention at clarity and correctness of the text. To avoid dubious statements like that on Page 2, Lines 10-11.

**Response: -** The authors would like to thank the reviewer for his valuable comments which helped us in reviewing the manuscript. As per the suggestion figures have been combined and few has been used as supplementary material. The writing style has been also improved and the manuscript has been checked thoroughly statement-by-statement.

Page 2 and line 10-11 has been revised.

**Change in the manuscript:** In the absence of appropriate region-specific models of wave propagation, ground motion prediction models are generally used to determine the hazard value.

**Comment 1: -** 1-17: tsunami

**Response: -** It has been changed in the revised manuscript.

**Comment 2: -** 1-18: Triggering tsunamis is nothing to do with ground shaking because tsunamis respond to residual, static deformation of the seabed, not to PGV or PGA.

**Response: -** Tsunami has been removed in the revised manuscript.

**Comment 3: -** 1-20: "subduction"

**Response: -** It has been changed in the revised manuscript.

**Comment 4: -** 1-20: I am not sure if you can call the India-Eurasia collision as "subduction zone" because the latter term commonly implies subduction of the oceanic lithosphere whereas in this case, we actually have continent-to-continent collision.

**Response: -** The word "subduction zone" has been replaced by "continent-to-continent collision".

**Change in the manuscript:** Besides, many great events (2015, Nepal earthquake) have originated from continental-to-continental collision.

**Comment 5: -** 2-6: Does aleatoric uncertainty include "randomness of ground motion prediction"? GMPE's are derived by people, not by nature. Maybe, better to say that it includes randomness of wave propagation and site amplification?

**Response: -** It has been changed as per the suggestion. The statement has been changed as follow

**Change in the manuscript:** One is due to randomness of the nature of earthquake, wave propagation, and site amplification named as aleatory uncertainty while other is due to incomplete knowledge of earthquake process named as epistemic uncertainty.

**Comment 6: -** 2-11: I do not see the logical connection between the sentence starting with "Generally, ground motion *: : :.*" and the next one. Logic tree is used to quantify all kinds of epistemic uncertainty, not only that related to GMPE's. Please consider re-formulating these paragraphs.

**Response: -** As per the suggestion this paragraph has been revised. It has been revised as follow

**Change in the manuscript:** Epistemic uncertainty is due to improper knowledge about the process involve in earthquake events and algorithms used to model them. Hence, in this study, logic tree framework has been used to reduce the epistemic uncertainty in the final hazard value calculation. In the absence of appropriate region-specific models of wave propagation, ground motion prediction models are generally used to determine the hazard value. The uncertainty in GMPEs can be reduced by incorporating logic tree in the hazard analysis study.

**Comment 7: -** 2-15: if weight is assigned, we cannot speak about "qualitative" assessment any more

**Response: -** This word has been removed in the revised manuscript.

**Comment 8: -** 2-21: "As per Bilham" – what is "per"?

**Response: -** "As per" has been replaced with "similar to"

**Comment 9: -** 2-28: "determined weighted mean"?

**Response: -** Apology for the typo. This statement has been revised as below

**Change in the manuscript:** Maximum magnitude has been determined using weighted mean considering three methods as increment factor on maximum observed magnitude, Kijko and Sellevoll (1989) and regional rupture characteristics (Anbazhagan et al. 2015b).

**Comment 10: -** 2-31: "viz." ?

**Response: -** "viz." has been replaced by "namely"

**Comment 11: -** 3-7: what is "SSA". Define explicitly before using abbreviation for the first time.

**Response: -** "SSA" is seismic study area and it has been mentioned in the revised manuscript.

**Comment 12: -** 3-8: an area cannot have only one single value of lon and lat. A point can, area – not.

**Response: -** The statement has been changed as follow

**Change in the manuscript:** The present study area has covered the longitude 84.6-85.65°E and latitude 25.2-25.8°N

**Comment 13: -** 3-10: give reference to Figure 1 in the beginning of Patna region description Figure 1: source labels not readable I suggest adding a supplementary table describing individual faults. Or, alternatively, to extend Table S1 with additional parameters like position, rupture length.

**Response: -** As per the suggestion, the refence of Figure 1 has been given in the beginning and Table S1 has been extended by providing the position (latitude and longitude of the end points), total fault length and rupture length.

**Comment 14: -** 3-16/17: redundancy

**Response: -** As per the suggestion the sentences are moved blow at relevant position.

**Comment 15: -** 3-28: this sentence looks redundant. The whole paragraph is better to move to the beginning of the current chapter.

**Response: -** As per the suggestion the whole paragraph is moved in the beginning of the paragraph.

**Change in the manuscript:** Based on damage distribution map i.e. isoseismal map (1833 Nepal earthquake and 1934 Bihar-Nepal earthquake) and location of Main Boundary Trust, Main Central Trust and Himalayan Frontal Thrust (HFT), a radius of 500 km has been selected for present SSA. The detail study about selecting SA of 500 km is given in Anbazhagan et al.

(2015a). Geographical information of India demonstrates that approximately 60 % of the land is highly susceptible to earthquakes (NDMA, 2010). The tectonic feature of SA has been compiled from the Seismotectonic Atlas (SEISAT, 2010) published by the Geological Survey of India (GSI, 2000). The seismotectonic map was developed by considering 500 km radius from Patna district boundary by considering linear sources (faults and lineaments) from SEISAT and published literatures (e.g. NDMA, 2010; Nath and Thingbaijam, 2012; Kumar et al., 2013). Separation of MBT and MCT has been done and all the faults along with MBT and MCT have also been numbered. Seismotectonic map for Patna District is shown in Figure 1. A brief description of seismicity and seismotectonics of SSA is given below.

**Comment 16: -** 4-21: it is still worth to provide GR-expression with 'a' and 'b' parameters Seismicity parameters 'a' and 'b' are discussed in both Sections 3.1 and 3.2. That is why present Section titles look somewhat misleading. Consider renaming these sections, for example, according to the derivation approach: period of completeness (3.1) vs magnitude of completeness (3.2).

**Response: -** Both the sections have been renamed as per the suggestion

**Comment 17: -** 5-13: why M4.5 was finally accepted as Mc? This statement comes into contradiction with following statements where Authors accept M6-model to be their reference model. M6 has different Mc values for the two regions.

**Response: -** Apology for the same. This statement has been removed as it's a typo error.

**Comment 18: -** General Remark to Section 3.2: Authors employ 9 different methods to estimate 'a', 'b', and Mc. But finally accept only one model, M6, giving the corresponding logic tree node weight = 0.5. That means all other models were given zero weights despite some of them (M1,3,5) show results similar to M6. Authors should clearer justify why they do neglect all other 8 models.

**Response: -** Nine methods have been used to check the variability in 'a', 'b', and Mc for the same study area. However as per Boomer et al. (2005) calculation effort increases dramatically with the inclusion of more branches in the logic tree. Therefore, Bommer et al. (2005) suggested avoiding using branches with slight differences between the options, in cases when those options result in very similar nodes. Hence only M6 has been used as M6 method is capable for $M_c$ calculation as it synthetically maximises the available data and stabilises the $M_c$ value.

**Change in the manuscript:** According to Boomer et al. (2005) calculation effort increases dramatically with the inclusion of more branches in the logic tree. Therefore, Bommer et al. (2005) suggested avoiding using branches with slight differences between the options, in cases when those options result in very similar nodes. Hence only M6 has been used as M6 method is capable for $M_c$ calculation as it synthetically maximises the available data and stabilises the $M_c$ value.

**Comment 19: -** 9-29: vulnerable?

**Response: -** Apology for the typo. This word has been replaced.

====== END ======

**Response to Reviewer 3**

**General Comment: -** Journal: NHESS Title: Probabilistic seismic hazard analysis using logic tree approach – Patna District (India) Author(s): Panjamani Anbazhagan et al. MS No.: nhess-2018-328 The article titled "Probabilistic seismic hazard analysis using logic tree approach Patna District (India)" utilize logic tree technique to conduct PSHA study for Patna District, India. Authors employ different branches in the logic tree for PSHA calculations to handle the epistemic uncertainties. Although the work is extensive, and the exerted efforts are great, this paper still needs many clarifications, so it can be accepted for publication. It is not well organized, and, in many parts, it is non-properly sequenced with non-threaded paragraphs, leaving the reader confused and suffering to catch the idea. The English language of the paper is poor and negatively affects the understanding of many paragraphs. English needs to be revised critically. Abbreviations should be mentioned at its first appearance. Avoid using the same abbreviation for two different terms (e.g. SA is used for spectral acceleration and for study area). What are SSA, MBT, MCT, S60,: ∴.etc. All abbreviations should be defined at their first appearance in the text. All localities, faults and geological structures mentioned in the manuscript should be shown on maps. I could not appropriately follow the seismotectonic part of the area due to lack of such illustrations.

**Response: -** The authors would like to thank the reviewer for his valuable comments which helped us in reviewing the manuscript. The manuscript has been revised thoroughly for English and flow has been maintained to make it easy for the readers. Abbreviations have been provided at the first place. SA is only used for the spectral acceleration in the revised manuscript. The faults mentioned in the manuscript has been shown properly and quality of the seismotectonic map has been improve.

**Introduction**

**Comment 1: -** Page 1, lines 20-21: Which gap? Please provide more explanation.

**Response: -** It is the Himalayan seismic gap and detail explanation is given in Bilham and Wallace (2005); which is also mentioned in the manuscript.

**Change in the manuscript:** The Himalayan seismic gap (Bilham and Wallace, 2005) and thick soft soil sediments makes the scenario more dangerous for cities close to Himalayan region.

**Comment 2: -** Page 2, lines 3-5: Very accurate sentence, but nothing is carried out in the end. Why this sentence is written here?

**Response: -** This sentence is mentioned to justify the need of the hazard analysis for the Patna city and in the present study an updated map, and methodology used to determine the hazard value at bedrock for Patna city.

**Comment 3: -** Page 2, line 27: I could not understand "Maximum magnitude has been determined weighted mean using increment *: : :: : :.*"

**Response: -** This statement has been revised and given below.

**Change in the manuscript:** Maximum magnitude has been determined using weighted mean considering three methods as increment factor on maximum observed magnitude, Kijko and Sellevoll (1989) and regional rupture characteristics (Anbazhagan et al. 2015b).

**Geology, Seismotectonics and seismicity of the study area (SA)**

**Comment 4: -** Page 3, line 8: coordinates here are for a point, it is not for an area.

**Response: -** The statement has been changed as follow

**Change in the manuscript:** The present study area has covered the longitude 84.6-85.65°E and latitude 25.2-25.8°N

**Comment 5: -** Page 3, line 29: "and published literatures" give references.

**Response: -** It has been mentioned in the revised manuscript.

**Change in the manuscript:** The seismotectonic map was developed by considering 500 km radius from Patna district boundary by considering linear sources (faults and lineaments) from SEISAT and published literatures (e.g. NDMA, 2010; Nath and Thingbaijam, 2012; Kumar et al., 2013).

**Comment 6: -** Page 4, lines 1-3: Authors should show the priority scheme in selecting the earthquake from each data base. I mean if the same earthquake is available in more than one database, which one will be selected? Which magnitude scale from which database has the first priority and which has the second and so on? Is the same magnitude scale for the same earthquake at different database yield the same value? All the above queries should be clarified in detail. Please show the start and end time of the catalogue to be able to assess its reliability.

**Response: -** The events have been selected from all the mentioned agencies. The duplicate events have been deleted and further the magnitude has been homogenized to moment magnitude scale. This is mentioned in the revised manuscript. Further the start and end time of the catalogue is also given in the revised manuscript.

**Change in the manuscript:** The events have been selected from all the mentioned agencies. The duplicate events have been deleted and further the magnitude has been homogenized to moment magnitude scale.

**Comment 7: -**Page 4, lines 15-18: Please revise the earthquake numbers in each magnitude range as their sum should be 818 as mentioned in Page 4 line 9.

**Response: -** Apology for the same. The correct number has been mentioned in the revised manuscript.

**a and b parameters**

**Comment 8: -**This is the most confusing part of the manuscript. In this section the a and b values are calculated for two regions (I and II). What is the role of these two areas and their seismicity parameters in the hazard calculations? The classical method used 178 seismic sources and the zoneless method used 7 area seismic zones. Why this is interfered in the current study. Secondly, the magnitude of completeness should be calculated before evaluating the seismicity parameters as GR parameters should use complete data only.

**Response: -** The seismic study area has been divided into two regions based on the seismicity. That is why a and b values are calculated for two regions (I and II). The hazard values are calculated using classical approach in which 178 seismic sources have been used as input parameter, whereas, in the zoneless approach, 7 areal sources have been used which are delineate based on the seismicity parameters.

a and b values have been calculated considering two ways one considering magnitude of completeness and other period of completeness.

**Comment 9: -** Magnitude of completeness Page 5, line 12: This great difference in the Mc values casts doubt on the calculated values. Please explain why different methods have such different outputs. Also justify the great difference in a and b values in lines 17-19. B values of 0.149 and 0.176 are not physically accepted. Again, it is not clear how the authors used the a and b values shown in this section in the hazard calculations?

**Response: -**We agreed with the reviewer, the difference in Mc values is due to the different algorithms used, which is also explained in the revised manuscript. However, we used these nine different methods to estimate the uncertainty in the seismicity parameters. The lower b-value is observed as it is calculated based on the magnitude of completeness, but it is not used for the analysis and is also explained in the revised manuscript.

**Change in the manuscript:** The lower b-value is observed as it is calculated based on the magnitude of completeness which may be due to the change in the algorithm as it selected the completed magnitude as minimum observed magnitude. This is not used further in the hazard calculation.

**Maximum magnitude estimation (Mmax)**

**Comment 10: -** Page 5, line 32: "based on b values" to add 0.5 based on b value, b value should range between 0.9 and -1.0, which is not the case here.

**Response: -**The calculated and adopted "b-values" is in the range of 0.8 to 1.0, hence as per the suggestion adding 0.5 to maximum magnitude observed is justifiable.

**Comment 11: -** The authors used the region-specific rupture technique to calculate Mmax and provide it the maximum weight. The technique depends on the ratio between the rupture length and the total fault length. My questions are: 1- Is the seismic record enough to be sure about the above ratio? The answer is NO as the authors themselves clarified when they justify the use of zoneless method, stating that "many sources given in Figure 1 are not well studied to prove its seismic activity". This raises great uncertainty on the maximum magnitude calculated for these seismic sources. 2- Is there any possibility to rupture the entire fault length in one earthquake? Recent studies suppose that the entire fault length will be ruptured in one earthquake when calculating the maximum earthquake.

**Response: -**We agreed with the reviewer but seismic sources we used are 178 in number which is enough as per our knowledge to justify the ratio and which can also be observed from the trend shown in Anbazhagan et al. (2015 a). However, in addition to that we also used other methods which is based on the seismicity of the region i.e. Kijko method and incremental method. All the sources used in the present study are from published literature and mentioned in the manuscript. There may be a possibility of total rupture of total fault length, however, as far as Himalayan seismotectonic is concerned, no study exists on this context as per knowledge. We may consider the total rupture in our future study.

**8.1 Classical approach**

**Comment 12: -** Page 9, line 27: Authors used 178 seismic sources. The seismicity of many of these faults are not well studied. It is not clear how the seismicity parameters are calculated for each single source. It is well known that GR model cannot be used to calculate a and b values for single faults. Slip rate could be used but with many not well studied sources, the results should be at least uncertain. Using logic tree does not mean ignoring use the right input parameters for each method.

**Response: -**We agreed with the reviewer that seismicity of the sources may not be properly studied, hence, due to that we used a well-defined approach explained by Anbazhagan et al. (2009). As far as this study is concerned, we did not calculate GR "a" and "b" parameter for single fault. Slip rate can be used but for determining the hazard value, we used well-defined algorithm defined by Cornell (1968), which does not require the same.

**Zoneless approach**

**Comment 13: -**Page 10, line27: use return period instead of "frequency of exceedance" Four models (figure 4) using zoneless approach (Frankel, 1995)

**Response: -**It has been replaced, as per the suggestion.

**Comment 14: -**Page 11, line 15: the return period 85 years (of what? This is most probably PGA)

**Response: -**Yes, it is the defined for PGA.

**Comment 15: -**Page 11, line 19: From which model the deaggregation plot is calculated? Or the authors used weighted deaggregation values based upon the weighs given for each of the four models. This should be very clear. Authors should explain why the results of the two methods are completely different in terms of hazard values and terms of the change in the spatial distribution (many low hazard areas in one method show very high hazard in the other method). This should be justified, as it is not enough to say for this the logic tree is created. A mistake could be done in the calculation or a method is not adequate for the region. Therefore, it is better to justify the use of zoneless methods.

**Response: -**The deaggregation has been calculated by considering the weighted mean from all the four models. This is mentioned in the revised manuscript. As these two methods have different input values, hence the results are different that is why logic tree approach has been used to reduce the uncertainty. The difference in results in explained in more details in the revised manuscript. The used of zoneless approach is due to spatial variability of the seismicity of the region and to estimate the hazard value where seismic source is not well studied. This is also explained in the revised manuscript.

**Change in the manuscript:** The deaggregation has been calculated by considering the weighted mean from all the four models.

**Comment 16: -**Page 12, line 5: Please add for 10% probability before "The PGA values" Final hazard map using logic tree

**Response: -**As per the suggestion, it has been added.

**Comment 17: -**Page 12, lines 26-27: As the high hazard values are related to the East and West Patna Fault, then, why the classical hazard values which are more related to the faults show very much less values?? Authors compared their results with previous studies. I recommend comparing the results of each method with the recent observations and with the previous studies to show a reason why the results are very inconsistent. If the current results are accurate, authors should recommend to change IS 1893 (2002) in Patna as the current hazard values highly exceed its summit.

Figure 1 is very unclear and need to be provided in a higher resolution way.

**Response: -**As per the results and calculations, PGA is higher near to the East and West Patna Fault (See Figure 8). As per the suggestions, the values form all the methods are also compared in the revised manuscript. Also detailed comparison with previous studies are revised in the revised manuscript

Figure 1 has been revised as per the suggestion and detailed source are given.

**Change in the manuscript:** It has seen from the mean deaggregation plot that the motion for 6.0 $M_w$ at 40 km hypocentral distance, 6.0 $M_w$ at 15 km hypocentral distance and 6.0 $M_w$ at 25.25 km hypocentral distance is predominant in case of Cornel's, Areal and Frankel's approach respectively considering 2 % probability in 50 years. However, the motion for 5.5 $M_w$ at 50 km hypocentral distance, 5.75 $M_w$ at 20 km hypocentral distance and 5.75 $M_w$ at 30.3 km hypocentral distance respectively predominant in case of Cornel's, Areal and Frankel's approach. The PGA values varies from 0.08 to 0.43 g, 0.29 to 0.41 g and 0.26 to 0.36 g in case of Cornel's, Areal and Frankel's approach respectively considering 2 % probability in 50 years. Whereas it from 0.04 g to 0.18 g, 0.09 g to 0.16 g and 0.09 g to 0.16 g respectively considering 10 % probability of exceedence in 50 years in case of Cornel's, Areal and Frankel's approach.

==== **END**====

---

## Author Response (AR2)

**Detailed Response Letter for NHESS-2018-328 Review Comments**

**Probabilistic seismic hazard analysis using logic tree approach- Patna District (India)**

Panjamani Anbazhagan[1], Ketan Bajaj[1], Karanpreet Matharu[1], Sayed S. R. Moustafa[2], Nassir S. N. Al-Arifi[2]

**Response to All the Reviewers Comments**

We thank all reviews and editor for their valuable to time to review manuscript and give valuable suggestions to improve the same. Most of the comments are suggestions to improve the current version of manuscript, which will incorporate in the revised manuscript. Few clarifications are requested, we have given our response for the same below and also highlighted that respective points will be added in the revised manuscript. As editor informed only prepare response to reviewers comments, we are not submitting revised manuscript now, But we have given revised text will be modified in the revised version.

**Response to Reviewer 1**

**Comment 1: -** The paper is well written with rich in technical analysis and content.

**Response: -** The authors would like to thank the reviewer for his/her valuable time for reviewing the manuscript and positive comments for the manuscript.

**Comment 2: -** The authors are requested to add more literature on previous seismic hazard studies. Some of the DSHA and PSHA studies done for entire India in the past is missing.

**Response: -** As per the suggestion, it has been added. **Please see page 1 and line number 29-30 in the revised with track changes.**

**Comment 3: -** Under the delineation of the seismic source model, authors missed the study "Characterization of Regional Seismic Source Zones in and around India" by Kolathayar and Sitharam (2012).

**Response: -** As per the suggestion, it has been added. **Please see page 8 and line number 1-2 in the revised with track changes.**

**Comment 4: -** Below sentence needs grammatical correction "178 seismic sources (shown in Figure 1 and given as Table ET1) have been used for determining the probability of occurrence of a specific magnitude, probability of hypocentral distance and probability of ground motion exceeding a specific value have been estimated as per Cornell (1968). "

**Response: -** As per the suggestion, it has been corrected as given below:

"178 seismic sources (shown in Figure 1 and given as Table ET1) have been used for determining the probability of occurrence of a specific magnitude, probability of hypocentral distance and probability of ground motion exceeding a specific value as per Cornell (1968)."

**Please see page 9 and line number 1-3 in the revised with track changes.**

**Comment 5: -** It is recommended to compare the GMPEs used: PGA for any constant magnitude with varying distance.

**Response: -** As per the suggestion, it has been added in the electronic supplement. **Please see page 7 and line number 14 in the revised with track changes.**

**Comment 6: -** 8.3 section heading may be renamed appropriately excluding Figure number in the bracket.

**Response: -** As per the suggestion, it has been modified.

**Comment 7: -** Please add a justification to use logic tree to integrate the hazard value obtained using different source models. Is it not advisable to take the maximum value than taking the average?

**Response: -** As per the suggestion, it has been added. Apology for the confusion, average and weighted average values are only used in the logic tree. **Please see page 13 and line number 17-20 & 32 in the revised with track changes**

**Comment 8: -** As the GMPE combination changes for different distance ranges, is it not advisable to take the maximum of PGA values than going for weights in logic tree, as for distance beyond 300 km, only single GMPE is used.

**Response: -** Apology for the confusion, but in the present study, weighted average has been used to determine the PGA using logic tree. It is corrected and made it clear in the revised version, **please see page 7 and line number 8 & 12 in the revised with track changes**

**Comment 9: -** Please do English language correction for the paper. For eg: "Whereas it from 0.04 g to 0.18 g, 0.09 g to 0.16 g and 0.09 g to 0.16 g respectively considering 10 % probability of exceedence in 50 years in case of Cornel's, Areal and Frankel's approach."

**Response: -** Apology, as per the suggestion, it has been modified.

**Comment 10: -** In Table 2 below GMPE is missing

Ramkrishnan, R., Sreevalsa, K., & Sitharam, T. G. (2019). Development of New Ground Motion Prediction Equation for the North and Central Himalayas Using Recorded Strong Motion Data. Journal of Earthquake Engineering, 1-24.

**Response: -** As per the suggestion, it has been added in Table 2.

**Comment 11: -**The section Zoneless approach talks about 7 areal zones. Is it not contradictory? Please make it clear whether Frankel approach was followed for different 7 zones separately or completely for the entire region? How both make a difference?

**Response: -** These zones were divided based on the seismicity parameters similar to Kolathayar and Sitharam (2012) and Vipin and Sitharam (2013). Frankel algorithm has been used for each of the seven regions separately to determine the hazard value. It is used to determine the difference between zoneless approach and Frankel's approach while diving the seismic sources differently. In the zoneless approach only one correlation distance i.e. 50 km has been used, whereas for Frankel approach two correlation distance i.e. $c = 50, 75$ km for model 1 and model 2 & 3.

**Response to Reviewer 2**

**General Comment: -** While the authors addressed all of my previous comments, two comments remain not addressed properly:

**Response: -** Apology for the same. These comments have been addressed in detail here.

**Comment 1: -** Comment 6: - Page 4: Authors gave unconvincing answer for the catalogue question, stating that "The events have been selected from all the mentioned agencies. Please read my question carefully and re-answer. I am talking in case of if a single earthquake is available in more than one database, which database will be selected (the best among them) and why?

**Response: -** Apology for the same, we agreed that the same earthquake event is presented with many agencies. On comparing it has been noticed that for most of the events the details are same, however if it isn't, we have take data from USGS.

**Comment 2: -** Comment 15: -Page 11, line 19: Authors should explain why the results of the two methods are completely different in terms of hazard values and terms of the change in the spatial distribution (many low hazard areas in one method show very high hazard in the other method). THIS SHOULD BE JUSTIFIED, as it is not enough to say for this the logic tree is created. A mistake could be done in the calculation or a method is not APROPRIATE for the region. Therefore, it is better to justify the use of zoneless methods.

**Response: -** In the zoneless approach, seven zones have been determined based on the spatial variability of the seismicity parameters. For determining the hazard value, the correlation distance of 50 kms has been used and the seismicity over each zone is similar and considering that as a source, hazard value has been determined. However, in four model Frankel's approach a

grid of size $0.02°\times0.02°$ is assumed as seismic source and two correlation distance i.e. $c = 50, 75$ km for model 1 and model 2 & 3.

[revised manuscript text omitted]